# CRMP4-mediated fornix development involves Semaphorin-3E signaling pathway

**Benoît Boulan[1†§], Charlotte Ravanello[1†], Amandine Peyrel[1], Christophe Bosc[1], Christian Delphin[1], Florence Appaix[1], Eric Denarier[1], Alexandra Kraut[2], Muriel Jacquier-Sarlin[1], Alyson Fournier[3], Annie Andrieux[1], Sylvie Gory-Fauré[1*‡], Jean-Christophe Deloulme[1*‡]**

[1]Univ. Grenoble Alpes, Inserm, U1216, CEA, Grenoble Institut Neurosciences, Grenoble, France; [2]Univ. Grenoble Alpes, Inserm, CEA, UMR BioSanté U1292, CNRS, CEA, Grenoble, France; [3]Department of Neurology and Neurosurgery, Montréal Neurological Institute, McGill University, Montréal, Canada

**\*For correspondence:**
sylvie.gory-faure@univ-grenoble-alpes.fr (SG-F);
Jean-Christophe.deloulme@univ-grenoble-alpes.fr (J-CD)

[†]These authors equally contributed to this study.
[‡]These authors also contributed equally to this work

**Present address:** [§]Cellular Neurobiology Research Unit, Institut de Recherches Cliniques de Montreal (IRCM), Montreal, Canada

**Competing interest:** The authors declare that no competing interests exist.

**ABSTRACT** Neurodevelopmental axonal pathfinding plays a central role in correct brain wiring and subsequent cognitive abilities. Within the growth cone, various intracellular effectors transduce axonal guidance signals by remodeling the cytoskeleton. Semaphorin-3E (Sema3E) is a guidance cue implicated in development of the fornix, a neuronal tract connecting the hippocampus to the hypothalamus. Microtubule-associated protein 6 (MAP6) has been shown to be involved in the Sema3E growth-promoting signaling pathway. In this study, we identified the collapsin response mediator protein 4 (CRMP4) as a MAP6 partner and a crucial effector in Sema3E growth-promoting activity. CRMP4-KO mice displayed abnormal fornix development reminiscent of that observed in Sema3E-KO mice. CRMP4 was shown to interact with the Sema3E tripartite receptor complex within detergent-*resistant* membrane (DRM) domains, and DRM domain integrity was required to transduce Sema3E signaling through the Akt/GSK3 pathway. Finally, we showed that the cytoskeleton-binding domain of CRMP4 is required for Sema3E's growth-promoting activity, suggesting that CRMP4 plays a role at the interface between Sema3E receptors, located in DRM domains, and the cytoskeleton network. As the fornix is affected in many psychiatric diseases, such as schizophrenia, our results provide new insights to better understand the neurodevelopmental components of these diseases.

## Editor's evaluation

In this work Boulan use a combination of biochemical investigations and in vivo mouse genetic analysis and colleague and describe how Sema3E affects the fornix development through CRMP4. The manuscript is of broad interest to investigators exploring the molecular mechanisms underlying circuit assembly in the mammalian central nervous system.

## Introduction

Axonal pathfinding and outgrowth are tightly regulated during central nervous system development, and allow neuronal tracts to be properly established. Axonal growth and guidance involve many signaling pathways triggered by various guidance cues such as semaphorins, netrins, and ephrins (*Kolodkin and Tessier-Lavigne, 2011*; *Raper and Mason, 2010*). These signals are transduced by transmembrane receptors to regulate intracellular effectors which then affect cytoskeleton dynamics,

leading to axonal outgrowth modulation. Dysregulation of these signaling cascades results in neuronal connectivity defects, and sometimes cognitive or psychiatric disorders. In this study, we focused on the signaling pathway mediated by semaphorin-3E (Sema3E) and its role in the development of the post-commissural fornix. The post-commissure of the fornix is the major output of the hippocampus and is mainly composed of subicular axons that primarily terminate in the mammillary bodies (*Witter, 2006*). The post-commissural fornix belongs to a set of limbic connections, including the Papez circuit, involved in memory, emotion, personality, and navigation (*Bubb et al., 2017*). Disruption of the fornix is associated with several psychiatric disorders and neurodegenerative diseases (*Koshiyama et al., 2020*; *Nowrangi and Rosenberg, 2015*).

The attractive effect of Sema3E on subicular neurons is crucial for fornix development and is enabled by a tripartite receptor complex composed of plexin-D1 (PlxD1), neuropilin-1 (Nrp1) and vascular endothelial growth factor receptor-2 (VEGFR2) (*Bellon et al., 2010*; *Chauvet et al., 2007*). Some receptors involved in semaphorin signaling pathways, such as Nrp1, have been shown to localize to detergent-resistant membrane (DRM) domains (including lipid rafts) (*Guirland et al., 2004*). Cholesterol- and sphingolipid-enriched DRM domains are involved in axonal growth and guidance, acting as platforms for ligand-induced receptor activation and recruitment of intracellular actors (*Simons and Sampaio, 2011*). For example, kinases downstream of semaphorin signaling, such as Akt or glycogen synthase kinase-3β (GSK3β), can be recruited to DRM domains where they are activated (*Gao and Zhang, 2009*; *Sui et al., 2006*). One of the major substrates of GSKβ is the collapsin response mediator protein family (CRMPs) (*Charrier et al., 2003*; *Hou, 2020*). The five CRMPs members have been implicated in numerous neuronal processes including neurite outgrowth (*Ji et al., 2014*; *Khazaei et al., 2015*), dendritic arborization (*Niisato et al., 2012*; *Niisato et al., 2013*), axonal regeneration (*Alabed et al., 2007*; *Alabed et al., 2010*; *Leung et al., 2002*), and axonal transport (*Kimura et al., 2005*; *Tsuboi et al., 2005*). Within this family, CRMP2 was the first to be implicated in Sema3A-dependent growth cone collapse in higher vertebrates (*Goshima et al., 1995*) and continues to be the most extensively studied. Sema3A stimulates CRMP2 phosphorylation through GSK3β activation which regulates the interaction between CRMP2 and the cytoskeleton (*Fukata et al., 2002*; *Khazaei et al., 2014*). Although repulsive Sema3A signaling has been well described in recent years, the molecules involved in Sema3E signaling and therefore in the growth of subicular neurons remain to be fully characterized. Recently, we showed that microtubule-associated protein 6 (MAP6) was crucial for fornix development due to its role in regulating axonal elongation induced by Sema3E (*Deloulme et al., 2015*). Here, we found that although MAP6 interacts with CRMP1, -2, and -4, only CRMP4 is required for Sema3E axonal outgrowth. In mice, knockout (KO) of *Crmp4* is associated with defective development of the post-commissural fornix. The results presented here show that CRMP4 binds the tripartite Sema3E-receptor complex in a DRM domain-dependent manner. Disruption of DRM domains blocks Sema3E-mediated axonal growth and affects CRMP4 phosphorylation by GSK3β. The integrity of DRM domains is thus critical for downstream activation of the PI3K/Akt/GSK3β pathway by Sema3E. Furthermore, we found that Sema3E signaling triggers a reduction of GSK3β-driven phosphorylation of CRMP4, and that the interaction of CRMP4 with the cytoskeleton is necessary to transduce Sema3E signaling.

Taken together, our results establish CRMP4 as a new specific downstream effector of the Sema3E signaling pathway, exerting its activity through DRM domains and phospho-regulated interactions with the cytoskeleton.

## Results

### CRMP1, -2, and -4 are MAP6 partners

MAP6 is crucial for Sema3E-dependent fornix development (*Deloulme et al., 2015*). To identify new effectors of the Sema3E signaling pathway, we performed MAP6-immunoaffinity chromatography experiments coupled with mass spectrometry-based protein identification. Protein extracts from adult brains were passed over affinity columns containing monoclonal antibody directed against the neuron-specific MAP6-N isoform. Bound proteins were then eluted by addition of an excess of MAP6 peptide containing the epitope targeted by the monoclonal antibody. Eluted fractions were analyzed by mass spectrometry to identify MAP6 partners. Interestingly, proteins in the CRMP family, members of which are known to play critical roles during semaphorin signaling (*Arbeille et al., 2015*;

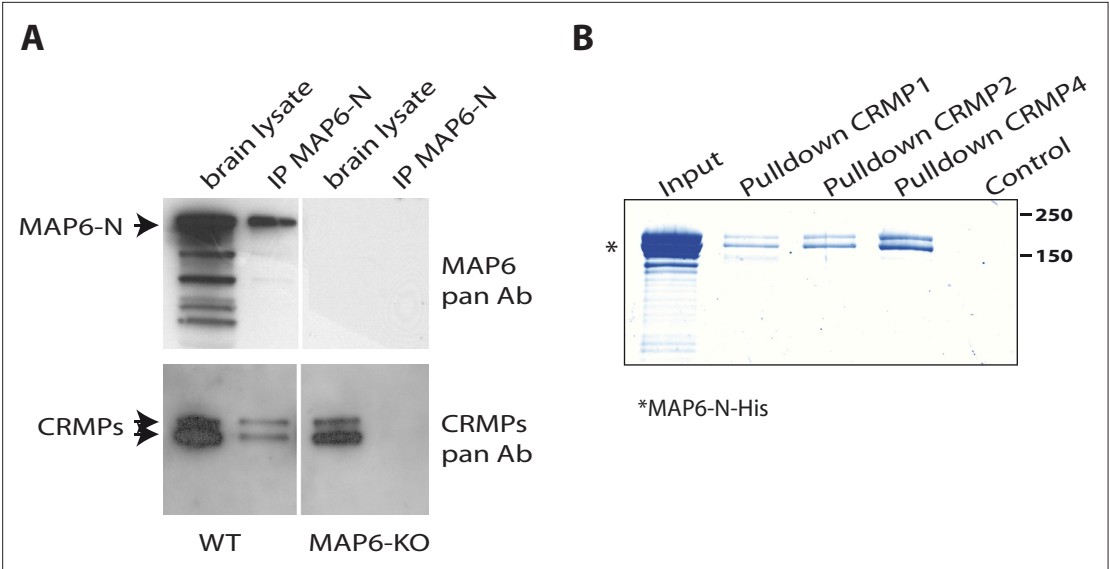

**Figure 1.** Identification of collapsin response mediator protein 1 (CRMP1), -2, and -4 as microtubule-associated protein 6 (MAP6)-binding partners. (**A**) Immunoprecipitation of endogenous CRMPs from adult mouse brain using mAb-175. CRMPs were revealed using anti-pan-CRMP antibody and MAP6 using anti-pan-MAP6 antibody. Lysate from MAP6-KO mouse brain was used as control. (**B**) Coomassie gel of pull-down experiments. Control Sepharose beads (control lane) or CNBr-coupled CRMPs-Sepharose beads were incubated with purified MAP6-N-His protein. Input lane corresponds to the total amount of MAP6-N-His (2 µg) incubated with the CRMPs-Sepharose beads (5 µL). Molecular weights are indicated in kDa.

The online version of this article includes the following source data and figure supplement(s) for figure 1:

**Source data 1.** Extended view of panels A and B.

**Figure supplement 1.** Recombinant collapsin response mediator protein family (CRMP) proteins.

---

*Goshima et al., 1995*; *Niisato et al., 2012*; *Quintremil et al., 2016*; *Uchida et al., 2009*; *Yamashita et al., 2007*), were identified as MAP6 partners. More precisely, we identified twelve specific peptides belonging to CRMP2, three specific peptides from CRMP4, two specific peptides from CRMP1, and six peptides shared by two or three of these CRMP proteins (*Supplementary file 1*). The presence and specificity of the CRMP-MAP6 complex was confirmed by co-immunoprecipitation experiments using wild-type (WT) and MAP6-KO brain lysates (*Figure 1A*). To determine whether MAP6 interacts directly with CRMPs, we performed pull-down experiments using sepharose beads coupled to recombinant CRMPs (*Figure 1—figure supplement 1*). Purified MAP6-N bound directly to CRMP1, CRMP2, and CRMP4 (*Figure 1B*).

## CRMP4 is crucial for Sema3E-dependent axonal outgrowth stimulation

To evaluate the function of MAP6-interacting CRMP isoforms in the axonal response to Sema3E, we used siRNAs to knock down CRMP1, CRMP2, or CRMP4 (*Figure 2A*). All siRNAs were effective, with inhibition ranging from 40% to 80% (*Figure 2A*). We then analyzed the effect of each siRNA on basal axonal growth compared to a control siRNA (*Figure 2B*). Decreased expression of CRMP2 weakly stimulated axonal growth (108%, p = 0.0275), knockdown of CRMP1 showed the same tendency but the increase was not significant (106%, p = 0.0976). In contrast, axonal-growth stimulation was more pronounced following inhibition of CRMP4 expression (116%, p = 0.0028) (*Figure 2B*). These results show that downregulation of CRMPs stimulated basal axonal growth in cultured subicular neurons.

To analyze the specific effect of Sema3E, axonal lengths of Sema3E-treated neurons were normalized relative to their respective control conditions (*Figure 2C and D*). As previously reported (*Chauvet et al., 2007*; *Deloulme et al., 2015*), application of Sema3E (5 nM) to WT cultured subicular neurons 48 hr after electroporation with control siRNA induced a 1.2-fold increase in axon length (*Figure 2C and D*, 120%, p < 0.0001). Although treatment with *Crmp1* or *Crmp2* siRNAs drastically reduced protein expression levels (*Figure 2A*), the Sema3E growth-promoting effect was still observed (*Figure 2D*, 119% and 124%, p = 0.0013 and p < 0.0001, respectively). Conversely, CRMP4 knockdown completely blocked the response to Sema3E (*Figure 2C and D*, 99%, p > 0.9999). These

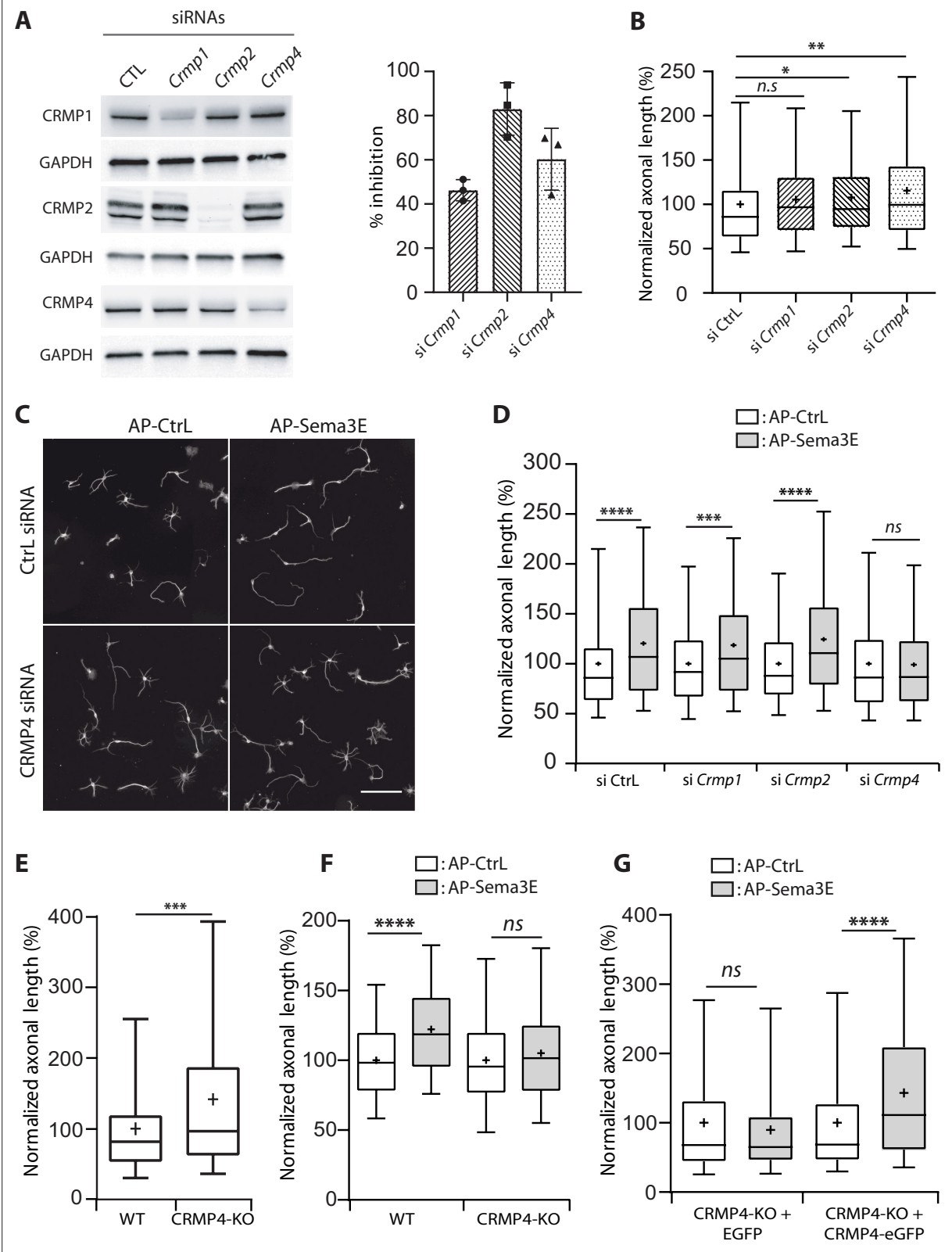

**Figure 2.** Effect of collapsin response mediator protein family (CRMP) downregulation on Semaphorin-3E (Sema3E) growth-promoting activity. (**A**) Western blot of crude extracts from cultured subicular neurons treated for 48 hr with siRNA for CRMPs or control siRNA, and blotted with CRMP and GAPDH antibodies. Right panel: quantification of siRNA efficiency, values are expressed as percent inhibition compared to control siRNA (n = 3 independent experiments). (**B**) Axonal length measured in cultured subicular neurons treated for 48 hr with control siRNAs and CRMP siRNAs (20 nM).

*Figure 2 continued on next page*

*Figure 2 continued*

Values were normalized to 100% for control siRNAs. (**C**) Representative images of subicular neurons electroporated with control or CRMP4 siRNA and cultured in the absence (AP-CtrL) or presence of 5 nM Sema3E (AP-Sema3E) for 48 hr. To measure axonal length, neurons were immunolabeled with anti-α-tubulin. Scale bar, 100 μm. (**D**) Quantification of the relative axonal length of cultured subicular neurons treated for 48 hr with 20 nM of control or CRMP siRNAs in the absence (AP-CtrL) or presence of 5 nM Sema3E (AP-Sema3E). Values were normalized to 100% for the AP control condition (AP-CtrL). (**E**) Quantification of relative axonal length in cultured wild-type (WT) and CRMP4-KO subicular neurons. Values were normalized to 100% for WT neurons. (**F**) Quantification of the relative axonal length of cultured WT and CRMP4-KO subicular neurons in the absence (AP-CtrL) or presence of 5 nM Sema3E (AP-Sema3E). Values were normalized to 100% for the AP control (AP-CtrL) condition. (**G**) Quantification of the relative axonal length of cultured CRMP4-KO subicular neurons electroporated with control plasmid coding for eGFP or for eGFP-tagged CRMP4 in the absence (AP-CtrL) or presence of 5 nM Sema3E (AP-Sema3E). Values were normalized to 100% for the AP control condition. In B, D, E, F, and G, crosses represent the mean, black bars represent the median, boxes represent the 25th and 75th percentiles, and the whiskers represent the 5th and 95th percentiles of values (72 neurons for each condition) from three independent cultures. Kruskal-Wallis non-parametric test followed by Dunn's multiple comparisons, ****p < 0.0001, ***p < 0.001, **p < 0.01, *p < 0.05 and n.s. p > 0.05.

The online version of this article includes the following source data for figure 2:

**Source data 1.** Extended view of panel A and raw data of A, B, D-G.

results show that among the three CRMPs isoforms interacting with MAP6, only CRMP4 is crucial for Sema3E's growth-promoting activity. The involvement of CRMP4 in the Sema3E signaling pathway was confirmed by experiments performed on neurons harvested from CRMP4-KO embryos. In accordance with the results from siRNA experiments (*Figure 2B*), CRMP4-KO neurons exhibited significantly longer axons (*Figure 2E*, 147%, p > 0.0004) and no longer responded to Sema3E stimulation (*Figure 2F*, 105%, p = 0.2652). Furthermore, electroporation of CRMP4-KO neurons with CRMP4-eGFP completely restored their response to Sema3E (*Figure 2G*, 143%, p < 0.0001).

Altogether, these results show an inhibitory role for CRMP isoforms on basal axonal growth and a specific mediation by CRMP4 of Sema3E-induced axonal outgrowth in cultured subicular neurons.

## CRMP4 is expressed by subicular neurons during post-commissural fornix formation

To assess whether CRMP4 can influence fornix formation through the Sema3E signaling pathway, we first determined the localization of CRMP4 during post-commissural fornix formation in the mouse brain. As reported in the rat brain (*Byk et al., 1998*), CRMP4 was predominantly expressed during embryonic and postnatal development between embryonic day 17.5 (E17.5) and postnatal day 7 (P7), and levels declined sharply after postnatal day 14 (P14) (*Figure 3—figure supplement 1*).

We next investigated the pattern of *Crmp4* gene expression in the developing brain using a *LacZ* reporter gene inserted into the *Crmp4* locus of CRMP4-KO mice (*Khazaei et al., 2014*). LacZ expression was mainly detected within thalamic regions, habenular nuclei, hypothalamic regions, inside the hippocampus including CA1, CA2, CA3, subiculum, and to a lesser extent in cortical areas of E17.5 brain slices.

Consistent with the expression pattern observed for the *Crmp4* gene, CRMP4 immunostaining was localized in neuronal tracts such as the anterior commissure, which is mostly comprised of cortical projections, or fornix projections (fimbria-fornix [fi] and the post-commissural fornix [pf]) expressing Nrp1, the Sema3E co-receptor (*Figure 3A* column a). The post-commissural fornix is mainly composed of subicular neuron projections (*Witter, 2006*). At higher magnifications, CRMP4 immunolabeling was detected both in the fimbria and the post-commissural fornix in Nrp1-positive tracts (*Figure 3A* columns c and d). This immunolabeling was specific since it was not observed on brain sections from CRMP4-KO mice (*Figure 3A*, columns b and e). In addition, cultured subicular neurons expressing the Nrp1 receptor also strongly expressed CRMP4 in the cell body, dendrites, axon, and growth cone (*Figure 3B*). Conventional confocal microscopy showed that CRMP4 and Nrp1 were strongly co-expressed, and co-localized in the growth cone in subicular neurons (*Figure 3B*, upper panels). Improved spatial resolution images of the growth cone were obtained using a confocal Airyscan detector (Zeiss) on two different acquisition planes. These images showed a close proximity between punctate Nrp1 and CRMP4 labeling positions (*Figure 3B*, lower panels). On both acquisition planes, the presence of yellow pixels suggests discrete co-localizations of the two proteins (*Figure 3B*, merge panels).

Taken together, our results indicate that during formation of the post-commissural fornix, CRMP4 is strongly expressed in subicular neurons and in their axonal projections in vivo.

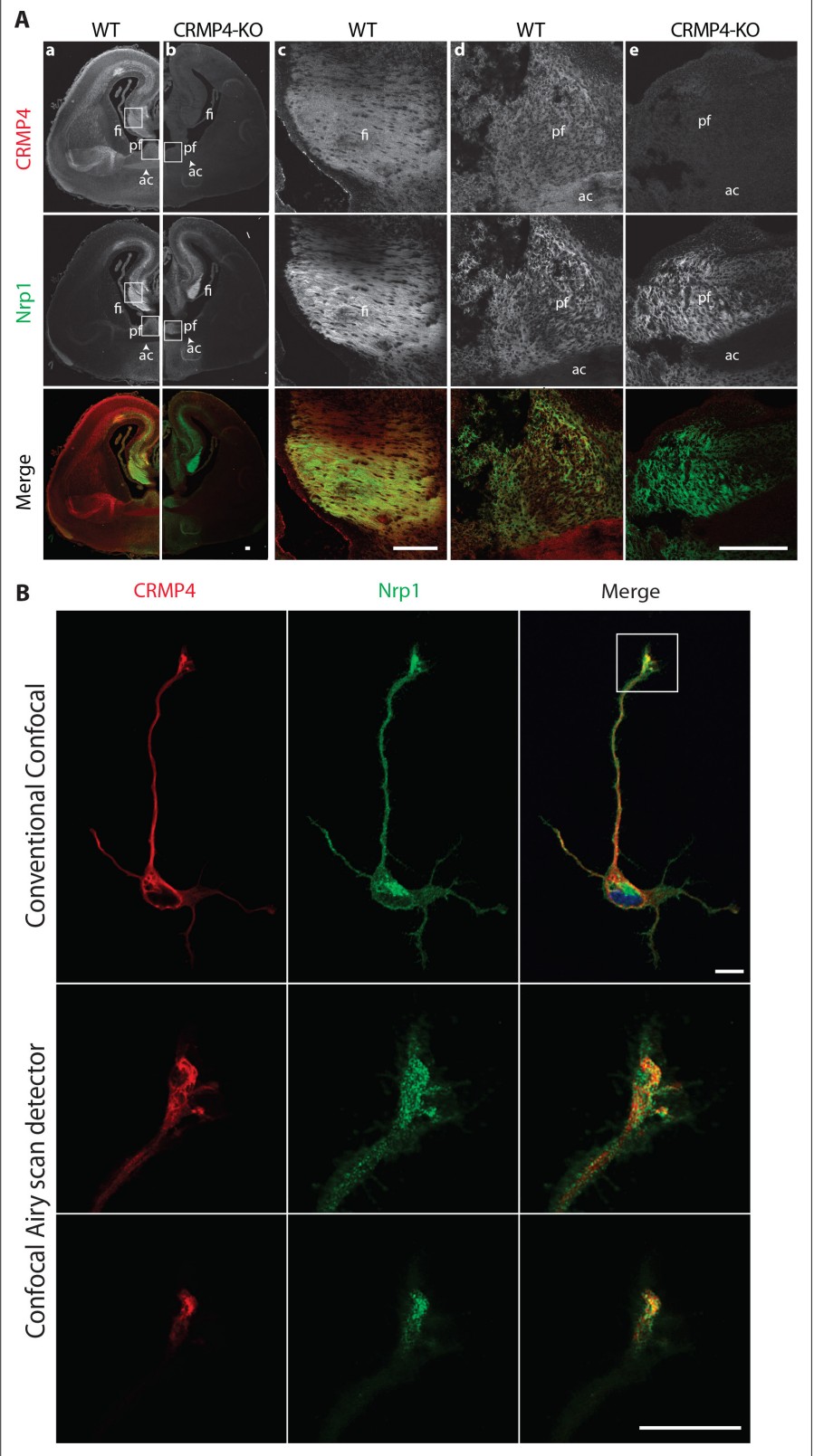

**Figure 3.** Collapsin response mediator protein 4 (CRMP4) expression in subicular neurons and fornix projections. (**A**) Coronal sections of embryonic day 17.5 (E17.5) wild-type (WT) (columns **a, c, d**) and CRMP4-KO (columns **b, e**) brain were immunolabeled with anti-CRMP4 and anti-Nrp1 antibodies. White squares (columns a and b) indicate the positions of fields of view shown in columns c–e. Abbreviations: ac: anterior commissure, fi: fimbria; pf: post-

*Figure 3 continued on next page*

*Figure 3 continued*

commissural fornix. Scale bars, 100 μm. (**B**) Cultured subicular neurons at 2 days in vitro (DIV) were immunolabeled with anti-CRMP4 and anti-Nrp1 antibodies. The conventional confocal plane shows the entire neuron. Images from the confocal Airyscan detail the growth cone on two serial acquisition planes. Scale bars, 10 μm.

The online version of this article includes the following source data and figure supplement(s) for figure 3:

**Figure supplement 1.** CRMP4 protein and gene expression during brain development.

**Figure supplement 1—source data 1.** Extended view of panel A.

## CRMP4 is required for fornix integrity

We then analyzed whether deletion of the *Crmp4* gene affects post-commissural fornix formation in vivo. To do so, we immunolabeled the post-commissural fornix using antibodies directed against Nrp1 on WT and CRMP4-KO brain sections from P0 littermate pups. At this developmental stage, the projections forming the post-commissural fornix pass behind the anterior commissure, passing through the anterior hypothalamus AH and terminating at the posterior hypothalamus (*Figure 4A*, sagittal diagram). To view the post-commissural fornix, we used a number of histological approaches. First, we prepared coronal sections at P0 and measured the diameter of the fornix at the level of the start of the anterior hypothalamic area (AHA) (*Figure 4A*, white arrowheads). We chose the AHA as an anatomical landmark due to its accuracy and its proximity to the post-commissural fornix, thus limiting bias due to differences in dorso-ventral positioning of brains during slicing. The diameter of the fornix was reduced by 53% in CRMP4-KO mice (*Figure 4A*, histograms, p = 0.0043). In contrast, the diameter of the neighboring mammillo-thalamic tract remained unchanged between WT and KO brains, indicating a specific role of CRMP4 in fornix formation (*Figure 4A*, histograms, p = 0.7922).

We then assessed the length of the entire post-commissural fornix and analyzed axon bundling patterns. To do so, we prepared sagittal slices with a cutting angle of 30° to visualize the entire post-commissural projection (*Deloulme et al., 2015*; *Figure 4B*, coronal diagram). The length of post-commissural fornices was measured using the anterior commissure as a starting point (*Figure 4B*, ac). The post-commissural fornix length was 26% shorter in CRMP4-KO brains (*Figure 4B*, histogram, p = 0.0173), appeared less fasciculated than in WT brains, and also presented divergent axons (*Figure 4B*, observed 3/6 KO, white arrow). Finally, to get a better quantitative estimation of the developmental defects affecting the post-commissural fornix in the absence of CRMP4, we used the tissue-clearing technique 3DISCO (*Belle et al., 2014*) after immunolabeling whole brain with Nrp1 antibodies. Brains were analyzed at three developmental stages: E18.5, P0, and P2. The post-commissural fornices were then manually segmented and 3D-reconstructed (*Figure 4C*, sagittal diagram and 3D reconstruction). The volume of post-commissural fornices in CRMP4-KO brains was smaller at all developmental stages than those in WT littermates' brains (*Figure 4C*). Furthermore, in CRMP4-KO brains, post-commissural fornices were less fasciculated, often containing some axon bundles deviating from the tract (*Figure 4C* white arrowheads, observed on 2/10 WT and 7/9 KO). Finally, quantification of the volume of the post-commissural fornix was significantly decreased – by 40% overall – in CRMP4-KO brains (*Figure 4C*, histogram, WT vs. KO p = 0.0070), with a 51% reduction recorded at P0 (*Figure 4C*, histogram, p = 0.0159). Altogether, these results clearly show that deletion of *Crmp4* affects both embryonic and postnatal development of the post-commissural fornix.

We then assessed if the developmental defect observed in CRMP4-KO mice could have consequences on fornix integrity in adulthood. To analyze post-commissural fornix anatomy in adults, we performed myelin staining using gold chloride on coronal sections at three rostro-caudal planes (*Figure 5B–D*) selected based on morphologic landmarks (*Figure 5A*, see Materials and methods section). In CRMP4-KO mice, the surface area occupied by the fibers in the post-commissural fornix progressively decreased in the rostro-caudal region, by 30%, 33%, and 47% (*Figure 5B, C, and D*, p = 0.0476, p = 0.0159, p = 0.0159, respectively) compared to the corresponding WT post-commissural fornix. Calculation of the fasciculation index also revealed that the defect in the fornix was more significant in its terminal part in CRMP4-KO brains (*Figure 5B, C and D*, –17%, –17%, –40%, p = 0.111, p = 0.111, p = 0.0317, respectively). As noted previously, no significant variation in mammillo-thalamic tract area was detected in adult brains, suggesting that the post-commissural fornix defect that appears during development persists into adulthood. Readers should note, the number of subicular neurons does not differ in adult WT- and CRMP4-KO brains (*Figure 5—figure supplement 1*).

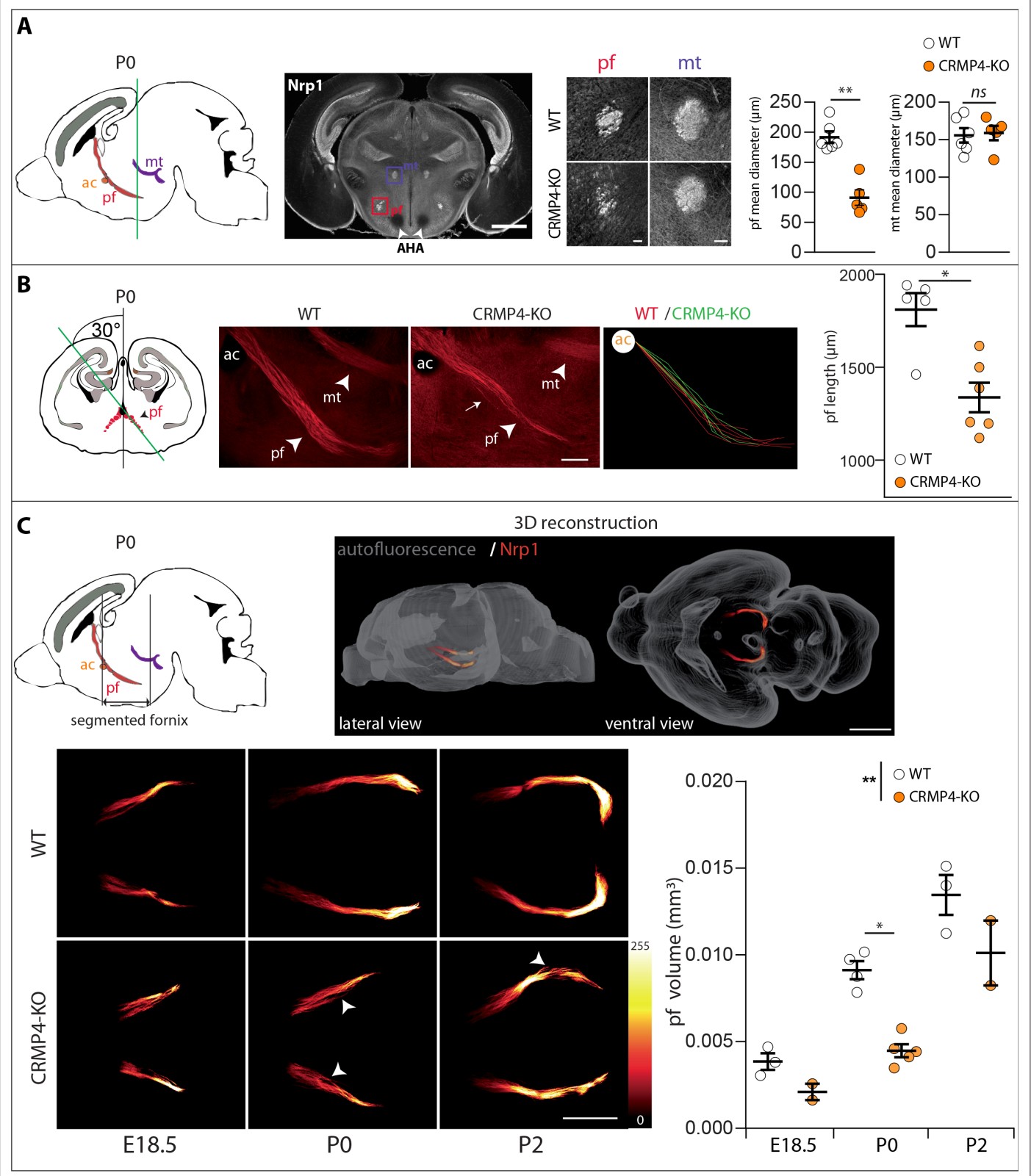

**Figure 4.** Effect of collapsin response mediator protein 4 knockout (CRMP4-KO) on post-commissural fornix formation. (**A**) From left to right: The sagittal diagram shows the projection stacks for the post-commissural fornix (pf; red) and mammillo-thalamic tract (mt, purple); the green line indicates the level of the coronal section. Representative coronal section corresponding to the onset of the anterior part of the anterior hypothalamic area (AHA) in postnatal day 0 (P0) wild-type (WT) and CRMP4-KO brains immunolabeled with anti-neuropilin-1 (Nrp1) antibody. Fornix and mammillo-thalamic tract

*Figure 4 continued on next page*

*Figure 4 continued*

were squared. Scale bar, 1 mm. Higher-magnification images show representative post-commissural and mammillo-thalamic tracts in CRMP4-KO and WT brains after Nrp1 immunolabeling. Scale bars, 100 µm. Dot plots show quantifications of both fornix and mammillo-thalamic tract diameters. Mean ± s.e.m. WT n = 6, KO n = 5, Mann-Whitney test, *p < 0.001. (**B**) From left to right: The coronal diagram shows the projection stack of post-commissural fornix (red) on a coronal plane. The green line indicates the plane of the sagittal sections. Representative sagittal sections with a cutting angle of 30° for P0 WT and CRMP4-KO brains were immunolabeled with anti-Nrp1 antibody. The lengths of post-commissural fornices were measured using ImageJ, taking the anterior commissure as the starting point. All traces measured from WT (n = 5; red) and CRMP4-KO (n = 6; green) were superimposed and are shown in the right panel. The dot plot shows quantifications of post-commissural fornices. Scale bar, 250 µm. Mean ± s.e.m. Mann-Whitney test, *p < 0.05. (**C**) The sagittal diagram shows the projection stack of the post-commissural fornix (pf, in red) and black lanes indicate the segmented portion of the fornix. The anterior commissure (ac, in orange) was used as the starting point to segment the fornix (upper left diagram). A representative P2 segmented fornix is shown in 3D reconstruction (Nrp1 in red) in the whole brain (autofluorescence in grey), illustrated in lateral view and ventral view (upper right panel). Reconstructions were produced using the ImageJ 3D Viewer plugin. Scale bar, 1 mm. Representative ventral z projections of WT and CRMP4-KO fornices at embryonic day 18.5 (E18.5), P0, and P2 are presented (bottom left panels). The fluorescence intensity of the segmented fornix is represented by a red-to-white color scale. White arrowheads point to axon bundles splitting off from the fornix. Scale bar, 500 µm. Histogram shows quantifications of the segmented post-commissural fornix volume at E18.5 (WT = 3; KO = 2), P0 (WT = 4, KO = 5) and P2 (WT = 3; KO = 2) for WT and CRMP4-KO mice. Bars indicate the mean volume ± s.e.m. Kruskal-Wallis non-parametric test followed by Dunn's multiple comparisons and one-way ANOVA to compare WT vs. KO, *p < 0.05, **p < 0.01.

The online version of this article includes the following source data for figure 4:

**Source data 1.** Raw data of panel A-C.

---

The fornix not only projects to the mammillary bodies through the post-commissural fornix (pf), but also projects into the septum (SPT) through the pre-commissural fornix, and into the anterior hypo-thalamus (AH) through the medial cortico-hypothalamic tract (mcht) (***Figure 5A***). Qualitative analysis of the pre-commissural fornix and mcht, using the hybrid uDISCO clearing method (see Materials and method section) on brains expressing YFP across the fornix system, revealed no obvious difference in these tracts between WT and CRMP4-KO mice. Analysis of the fornix at the entry to mammillary bodies confirmed the loss of fibers in KO mice (***Figure 5—figure supplement 2***). These observations suggest that the post-commissural projection is the main fornix projection altered in CRMP4-KO.

## Crmp4 gene shows a genetic interaction with Sema3e involved in post-commissural fornix development

The defect in fornix development combined with defasciculation observed in CRMP4-KO mice pheno-typically copies the defects observed in Sema3E-KO mice (***Chauvet et al., 2007***).

To investigate genetic interactions between *Crmp4* and *Sema3e* during fornix development in vivo, we analyzed trans-heterozygous animals. The post-commissural fornix was segmented from the anterior commissure in whole brains cleared by iDISCO at P0 (***Figure 6***; ***Renier et al., 2014***). The fornix volume was 31% decreased in *Crmp4*$^{+/-}$/*Sema3e*$^{+/-}$ double mutants compared to WT controls (***Figure 6A and B***, p = 0.0061). No significant decrease was observed in *Crmp4*$^{+/-}$ or *Sema3e*$^{+/-}$ single mutants (***Figure 6B***, –4% and +2%, p > 0.999 for both). As a control, the mammillo-thalamic volume showed no variation (***Figure 6C***). The fornix defects found in single-gene KO and in double hetero-zygous animals, but not in single heterozygous mice, are strong indicators of an in vivo interaction between *Crmp4* and *Sema3e* during fornix development.

## CRMP4 interacts with the Sema3E receptor complex

Sema3E signaling is triggered by the tripartite complex comprised of PlxD1 (which binds Sema3E), its co-receptor Nrp1, and VEGFR2 which allows signal transduction leading to axonal growth (***Bellon et al., 2010***). Previous studies have shown that CRMPs can associate with plexin-A receptors to trans-duce class 3 semaphorin signaling (***Deo et al., 2004***; ***Jiang et al., 2020***; ***Schmidt et al., 2008***; ***Sekine et al., 2019***). We therefore investigated a possible interaction between Sema3E receptors and CRMP4. To do so, ectopic PlxD1, Nrp1, and VEGFR2 were simultaneously expressed in HEK293T/17 cells in combination with CRMP4. We detected PlxD1, Nrp1, and CRMP4 proteins in VEGFR2 immunoprecip-itates, indicating the presence of a complex involving both Sema3E receptor components and CRMP4 (***Figure 7A***).

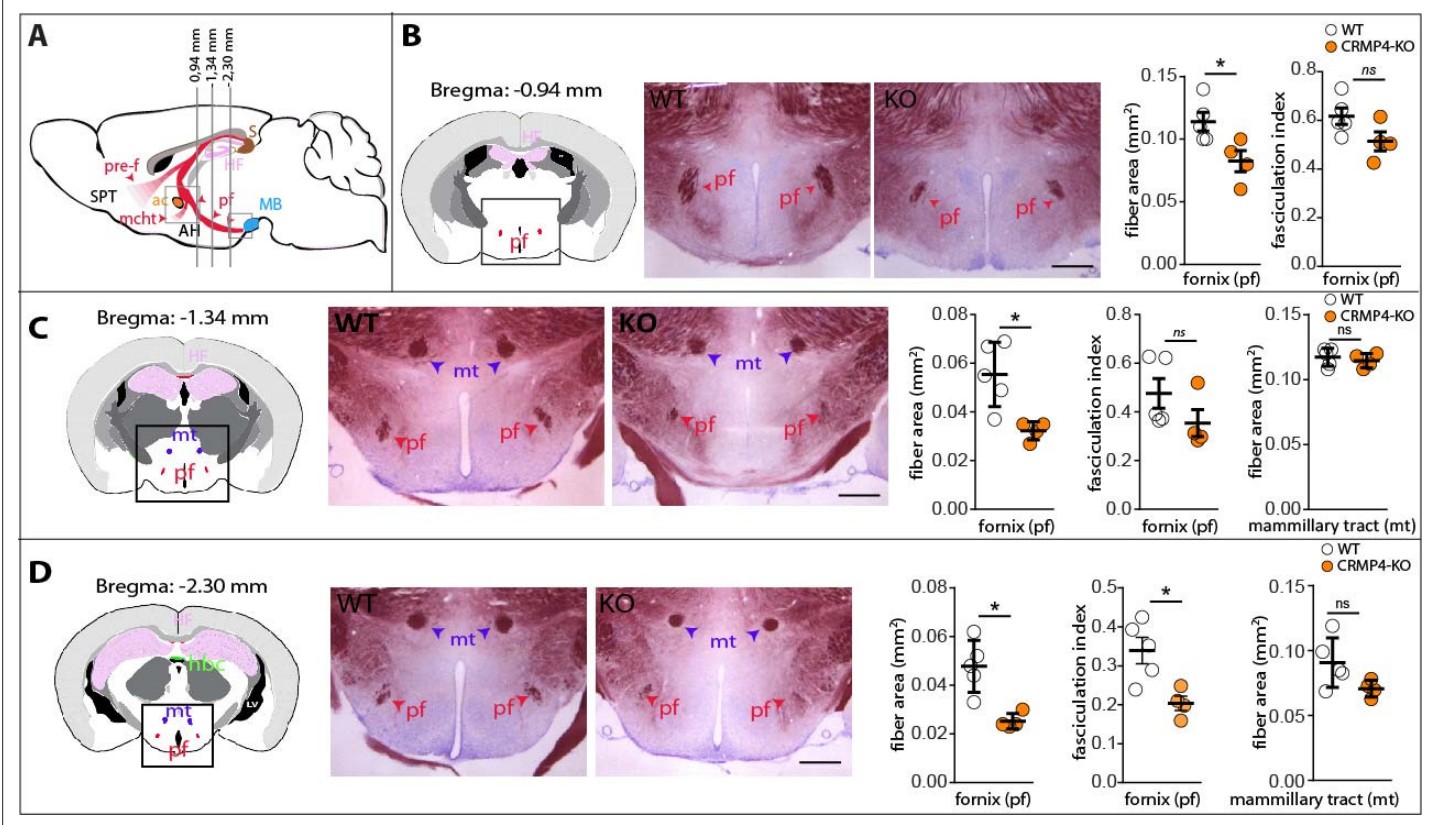

**Figure 5.** Deletion of collapsin response mediator protein 4 (CRMP4) affects fornix integrity in the adult brain. (**A**) Schematic representation of the fornix tract (red region) on a sagittal diagram. Grey lines indicate the Bregma positions, in mm, of the coronal planes presented in B, C, and D. (**B–D**) Coronal brain sections from adult wild-type (WT) and CRMP4-knockout (KO) brains were stained with gold chloride. Diagrams of adult brain coronal slices illustrating the anatomical levels selected to measure surfaces covered by the fornix (pf) and mammillo-thalamic tract (mt). Square in the diagrams indicate the regions shown on the coronal sections stained with gold chloride. Histograms show quantifications of fiber areas and fasciculation index, mean ± s.e.m. WT = 5, KO = 4. Mann-Whitney test, *p < 0.05. Scale bar, 250 μm. Abbreviations: HF: hippocampal formation; MB: mammillary body; AH: anterior hypothalamus; SPT: septum; pre-f: pre-commissural fornix; pf: post-commissural fornix; mt: mammillo-thalamic tract; mcht, medial cortico-hypothalamic tract; ac: anterior commissure; S: subiculum.

The online version of this article includes the following source data and figure supplement(s) for figure 5:

**Source data 1.** Raw data of panels A-D.

**Figure supplement 1.** Comparison of the number of subicular neurons in wild-type (WT) and collapsin response mediator protein 4 (CRMP4) dorsal subicula.

**Figure supplement 2.** Visualization of the whole fornix system by uDISCO tissue clearing.

In further experiments, ectopic PlxD1, Nrp1, and VEGFR2 were individually expressed in HEK293T/17 cells in combination with CRMP4-tGFP. VEGFR2 alone did not co-immunoprecipitate with CRMP4-tGFP (*Figure 7B*), whereas isolated Nrp1 or PlxD1 did (*Figure 7C and D*). These results strongly suggest that CRMP4 forms a complex with the Sema3E tripartite receptors through interactions with both PlxD1 and Nrp1.

## CRMP4/Sema3E receptor complexes are located in DRM domains

The concentrations of salt (up to 500 mM NaCl) and detergent (0.5% Triton X-100) used in our immunoprecipitations were relatively high, thus the interactions between CRMP4 and Sema3E receptors were robust. Indeed, these interactions were even resistant to concentrations of up to 1% Triton X-100 (*Figure 8A*), suggesting a possible role of DRM domains in their formation. To test this possibility, HEK293T/17 cell lysate in 1% Triton X-100 buffer and expressing PlxD1, Nrp1, and CRMP4 were subjected to sucrose gradients allowing separation of cellular constituents based on their density (*Figure 8B*). Endogenous flotillin, a protein enriched in DRM domains, was used as a marker of these

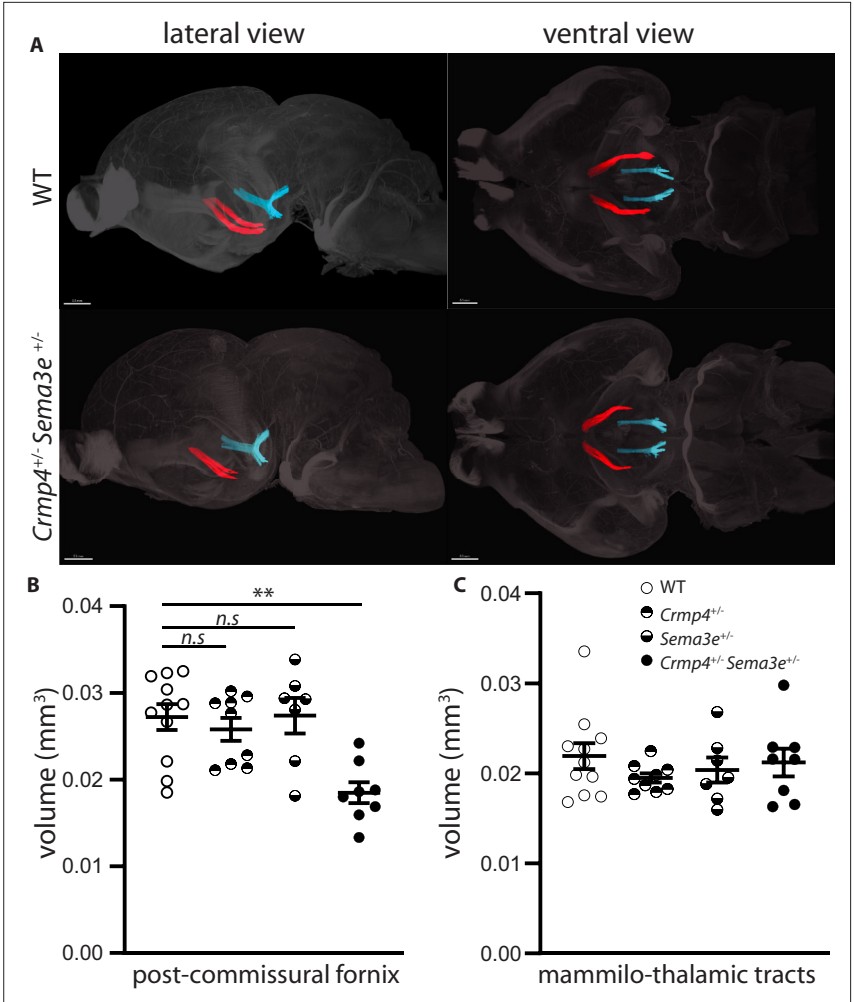

**Figure 6.** Post-commissural fornices are affected in collapsin response mediator protein 4 (*Crmp4*$^{+/-}$)/Semaphorin-3E (*Sema3*e$^{+/-}$) double heterozygotes. (**A**) Postnatal day 0 (P0) segmented fornices (red) and mammillo-thalamic tracts (blue) 3D reconstructions in whole brains are illustrated in lateral view (left panel) and ventral view (right panel). Whole brains from single and double heterozygotes for *Crmp4* and *Sema3e* and their wild-type (WT) littermates at P0 were immunolabeled with Nrp1 antibody and cleared according to the iDISCO protocol. Fornices and mammillo-thalamic fiber bundles were segmented using Imaris software. (**B**) Quantification of fornix (left panel) and mammillo-thalamic tract (right panel) volumes in P0 brains from WT (n = 11), *Crmp4*$^{+/-}$ (n = 9), *Sema3e*$^{+/-}$ (n = 7), and *Crmp4*$^{+/-}$/*Sema3e*$^{+/-}$ (n = 8) mice. Voxel size: x = 3.02; y = 3.02; z = 3 µm; Bars indicate the mean volume ± s.e.m. Kruskal-Wallis non-parametric test followed by Dunn's multiple comparisons, **p < 0.01, n.s. p > 0.05.

The online version of this article includes the following source data for figure 6:

**Source data 1.** Raw data of panel B.

---

structures to ensure successful fractionation of the membranes. Flotillin was mainly detected at the 5–38% sucrose interface, in the fractions corresponding to the DRM (fractions 3, 4, and 5), as well as in the fractions consisting of the remaining plasma membrane (RPM, fractions 10, 11, and 12) (*Figure 8B*). CRMP4, Nrp1, and PlxD1 were also found in the DRM fractions (PlxD1: 7.7%, Nrp1: 4.3%, CRMP4: 1.6%, flotillin: 53.2%, n = 3) but in much lower amounts compared to the membrane fractions (PlxD1: 87.6%, Nrp1: 90.4%, CRMP4: 90.0%, flotillin: 32.4%, n = 3), suggesting that only a small proportion of the Sema3E receptors and CRMP4 locates in DRM (*Figure 8B*).

To confirm the presence of receptors and CRMP4 in DRM, HEK cells were treated with methyl-beta-cyclodextrin (MβCD), a cholesterol-depleting agent that can disrupt DRM (*Gimpl and Gehrig-Burger, 2007*; *Kilsdonk et al., 1995*). Treating cell cultures with 10 mM MβCD for 7 hr resulted in a complete elimination of CRMP4-containing and receptor-containing DRM, and partial reduction

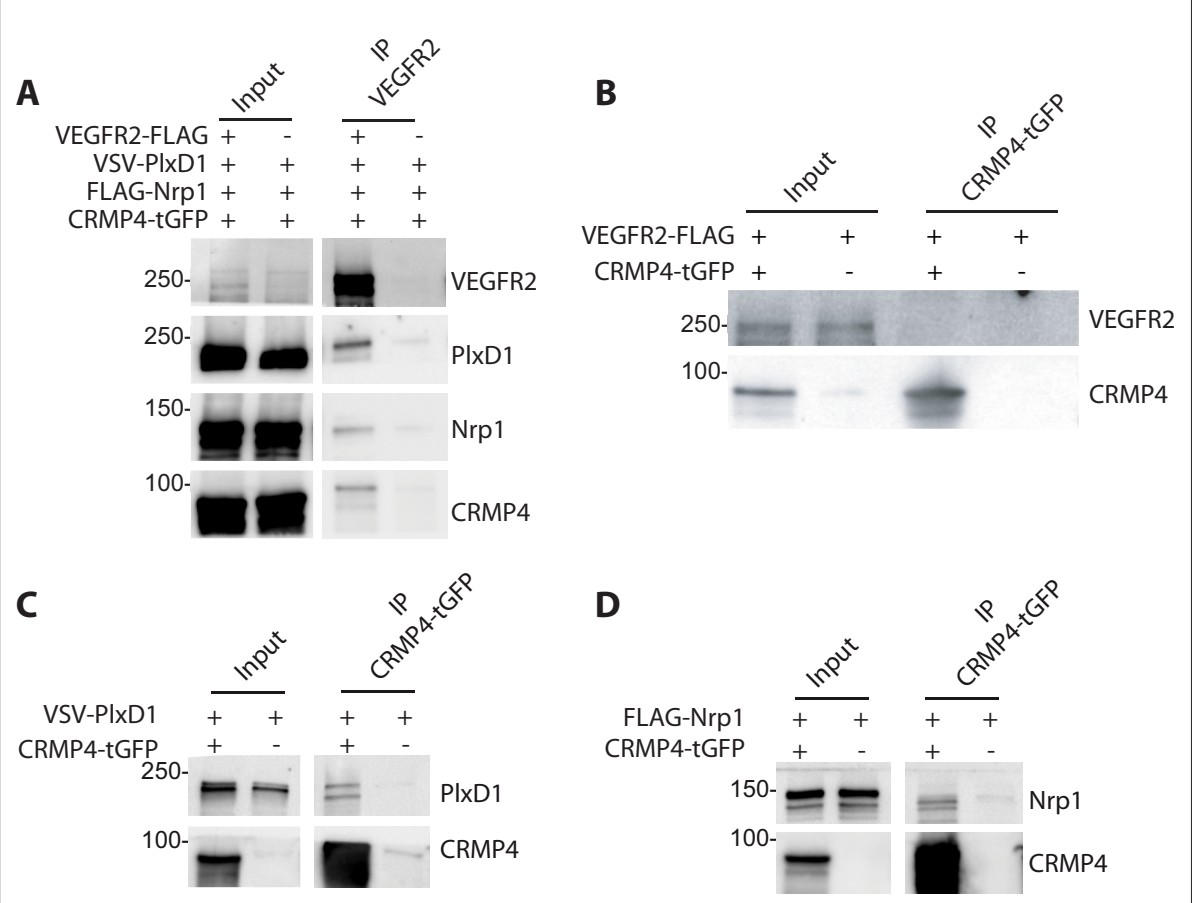

**Figure 7.** Collapsin response mediator protein 4 (CRMP4) forms complexes with Semaphorin-3E (Sema3E) receptors. (**A**) Western blot showing co-immunoprecipitation of CRMP4 and Sema3E receptors. HEK293T/17 cells were transfected with plasmids encoding CRMP4-tGFP, VEGFR2-FLAG, VSV-PlxD1, and FLAG-Nrp1 proteins. Cells were lysed in buffer containing 0.5% Triton X-100, and immunoprecipitation was performed using a polyclonal anti-VEGFR2 antibody. (**B–D**) Western blot showing co-immunoprecipitation of CRMP4 with each individual Sema3E receptor. HEK293T/17 cells were transfected with plasmids encoding CRMP4-tGFP and with plasmids encoding VEGFR2-FLAG (**B**), VSV-PlxD1 (**C**), or FLAG-Nrp1 (**D**). Cells were lysed in buffer containing 0.5% Triton X-100 and immunoprecipitation was performed using a polyclonal anti-tGFP antibody.

The online version of this article includes the following source data for figure 7:

**Source data 1.** Extended view of panels A-D.

of flotillin-containing DRM (*Figure 8C*). This difference in efficacy between flotillin-containing and Sema3E receptor-containing DRM is expected, and may reflect heterogeneity across DRM domains (*Foster et al., 2003*). To understand how CRMP4-Sema3E receptor complexes are distributed within DRM and membrane fractions, we analyzed co-immunoprecipitations of Sema3E receptors with CRMP4 in pooled DRM fractions (fractions 3, 4, and 5) or in pooled RPM fractions (fractions 10, 11, and 12) (*Figure 8D*). Surprisingly, even though the two receptors co-immunoprecipitated with CRMP4 in both RPM and DRM fractions, the ratio of receptors present in the immunoprecipitation relative to the amount of immunoprecipitated CRMP4 was considerably higher in the DRM fraction than in the RPM fraction (Nrp1 and PlxD1 signals represent 62% and 25% of CRMP4 signal, respectively, in DRM fractions, as compared to 0.18% and 0.69% of CRMP4 signal, respectively, in RPM fractions).

Altogether these results strongly indicate that the complexes between CRMP4 and Sema3E receptors are mainly found within DRM domains in cells.

## DRM domain integrity is required for the axonal growth-stimulating activity of Sema3E

DRM are dynamic membrane domains formed from the assembly of cholesterol, sphingolipids, and proteins that can be stabilized and coalesce to function as a scaffold for cell signaling complexes

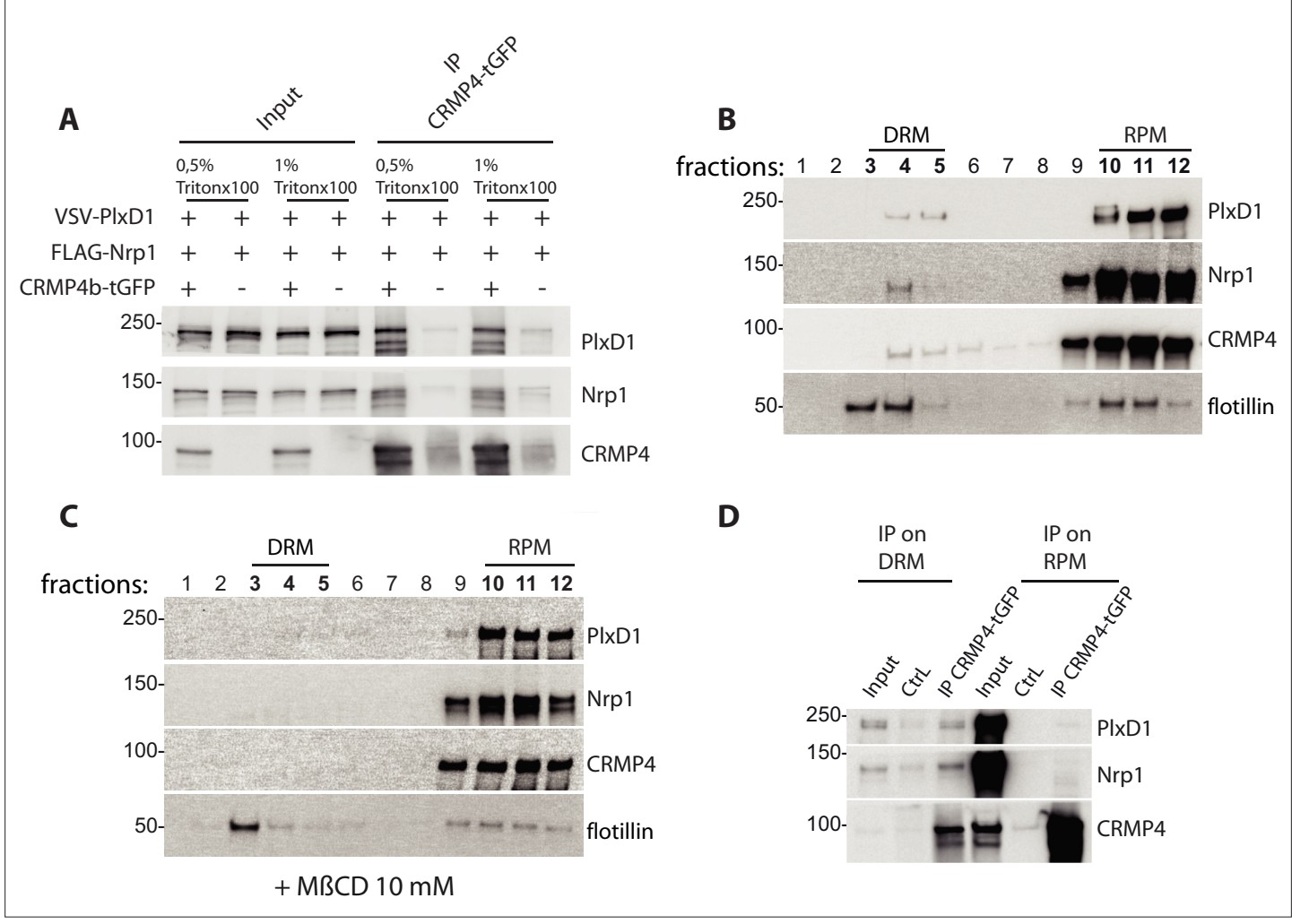

**Figure 8.** Collapsin response mediator protein 4 (CRMP4) interacts with the Semaphorin-3E (Sema3E) receptor complex in detergent-resistant membrane (DRM) fractions. (**A**) HEK293T/17 cells were transfected with plasmids encoding CRMP4-tGFP, VSV-PlxD1, and FLAG-Nrp1 proteins. Cells were lysed in buffer containing 0.5% Triton X-100 (as used in **Figure 7**) or 1% Triton X-100. Immunoprecipitations were performed using a polyclonal anti-tGFP antibody. Proteins were analyzed by western blotting with PlxD1, Nrp1, and tGFP antibodies. (**B and C**) Sucrose-density gradient fractionation of lysates prepared with 1% Triton X-100 from HEK293T/17 cells transfected with plasmids encoding ectopic CRMP4-tGFP, VSV-PlxD1, and FLAG-Nrp1 proteins. HEK293T/17 cells were cultured in the absence (**B**) or presence (**C**) of 10 mM methyl-beta-cyclodextrin (MβCD) for 7 hr. Fractions were then analyzed by western blot. Ectopic proteins were assayed with PlxD1, Nrp1, and tGFP antibodies, and endogenous flotillin was detected with flotillin antibodies. (**D**) Western blot showing co-immunoprecipitations between CRMP4 and the Sema3E receptor complex from sucrose gradient fractions corresponding to DRM (3, 4, and 5) or to the remaining plasma membrane (RPM) (10, 11, and 12, see left panel in A), using a polyclonal anti-tGFP antibody. In control conditions (CtrL), no tGFP antibody was added to the pooled fractions. Equivalent volumes of pooled fraction (DRM and RPM) were loaded (input).

The online version of this article includes the following source data for figure 8:

**Source data 1.** Extended view of panels A-D.

(**Lingwood and Simons, 2010**). These domains are involved in the mediating signaling through several molecules controlling growth and axonal guidance (**Averaimo et al., 2016**; **Davy et al., 1999**; **Guirland et al., 2004**; **Hérincs et al., 2005**). To determine whether DRM domains are essential for Sema3E signaling, we treated subicular neurons with increasing doses of MβCD and assessed the effect of the resulting DRM disruption on Sema3E-induced outgrowth (**Figure 9**).

We first investigated the effect of MβCD on axonal growth, and found, in accordance with previous work (**Ko et al., 2005**), that the addition of 1 or 1.5 mM of MβCD had a very mild effect, resulting in a 1.1-fold increase in axonal length (**Figure 9—figure supplement 1**, 112% of control for 1 mM MβCD, and 115% for 1.5 mM). Next, Sema3E, in the absence of MβCD, induced an expected 1.3-fold

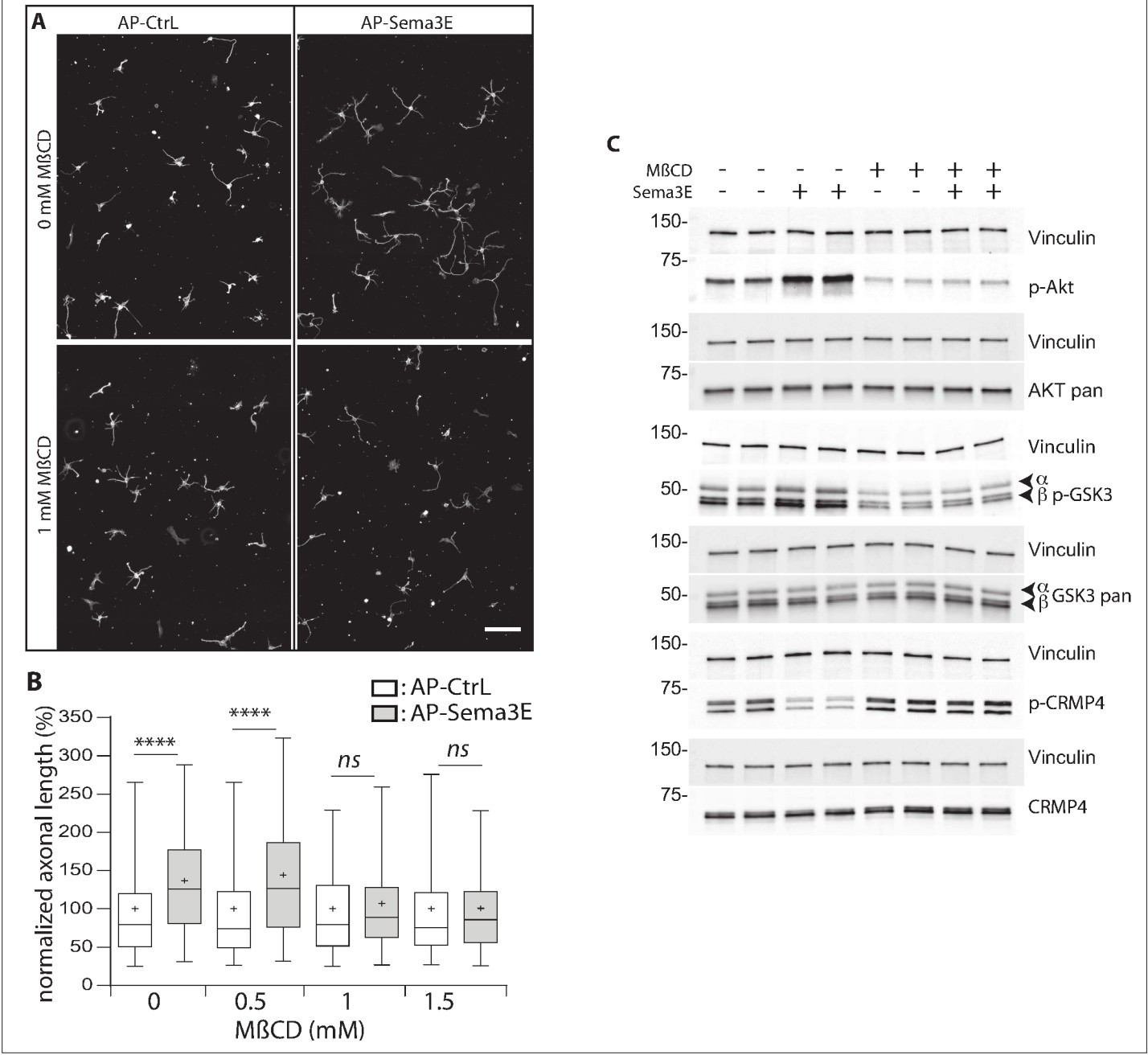

**Figure 9.** Methyl-beta-cyclodextrin (MβCD) inhibits the downstream Semaphorin-3E (Sema3E) signaling pathway. (**A**) Representative images of wild-type (WT) subicular neurons cultured in the absence (AP-CtrL) or presence of 5 nM Sema3E (AP-Sema3E) for 48 hr, with or without MβCD pretreatment (1.5 mM). Cells were fixed and stained with an anti-tubulin antibody to visualize neuronal extensions. Scale bar 100 μm. (**B**) Quantification of relative axonal lengths in WT subicular neurons cultured in the absence (AP-CtrL) or presence of 5 nM recombinant Sema3E (AP-Sema3E) for 48 hr, following a 2 hr pretreatment with different MβCD concentrations (from 0 to 1.5 mM). Crosses represent the mean, black bars represent the median, boxes represent the 25th and 75th percentiles, and the whiskers represent the 5th and 95th percentiles of values (300 neurons for each condition from two independent cultures). Values were normalized to 100% for control conditions (AP-CtrL). Kruskal-Wallis non-parametric test followed by Dunn's multiple comparisons, ****p < 0.0001 and n.s. p > 0.05. (**C**) Representative western blots of subicular neurons exposed or not to 4 mM MβCD for 30 min, followed or not with 5 nM recombinant Sema3E for 11 min. Proteins and phosphorylated proteins were assessed using their corresponding antibodies. n = 2 independent experiments, samples were loaded in duplicate.

The online version of this article includes the following source data and figure supplement(s) for figure 9:

**Source data 1.** Extended view of panel B and raw data of C.

**Figure supplement 1.** Effect of methyl-beta-cyclodextrin (MβCD) on axonal growth of subicular neurons.

increase in axonal length (*Figure 9A and B*, 137% of control, p < 0.0001), whereas in the presence of 1.0 and 1.5 mM MβCD, Sema3E no longer stimulated axonal growth (*Figure 9A and B*, 107% and 101% of control, p = 0.1150 and p > 0.9999, respectively). These results strongly suggest that DRM domain integrity is required for Sema3E signaling.

To further assess this possibility, we investigated the downstream signaling cascade induced by applying Sema3E to subicular neurons. Sema3E stimulates axon growth through activation of the phosphatidylinositol 3-kinase (PI3K)/Akt/GSK3β pathway (*Bellon et al., 2010*). By binding to PlxD1, Sema3E activates VEGFR2 which triggers the recruitment and activation of PI3K. Activated PI3K phosphorylates downstream Akt, which in turn phosphorylates GSK3β to inactivate it. CRMPs including CRMP4 are well-known substrates of GSK3β, which catalyzes their phosphorylation on threonine 509 (*Cole et al., 2006*; *Cole et al., 2004*). We thus determined whether CRMP4 phosphorylation was modified downstream of Sema3E signaling. To do so, we treated starved subicular neurons with Sema3E for 11 min and monitored the phosphorylation status of the downstream effectors by western blot using appropriate antibodies (against Akt p-Ser473, GSK3β p-Ser21/9, and CRMP4 p-Thr509) (*Figure 9C*). As expected, Sema3E treatment triggered the signaling cascade with an increase in Akt and GSK3β phosphorylation (203% ± 35% and 153% ± 9.5 of the control, respectively). In the same experiments, a considerable decrease in CRMP4 phosphorylation (43% ± 5.4 of the control) was observed. This decrease strongly suggests that the phosphorylation state of CRMP4 is regulated by Sema3E signaling.

We next investigated whether the Sema3E signaling cascade depends on DRM integrity. Subicular neurons were once again treated with Sema3E for 11 min in the presence of MβCD. Depleting cholesterol using MβCD (*Ohtsuka et al., 2017*; *Raghu et al., 2010*; *Scheinman et al., 2013*) inhibits the basal PI3k/AKT pathway (46% ± 15% and 65% ± 12 of control signal for p-AKT and p-GSK3β, respectively). In the presence of MβCD, the application of Sema3E does not lead to increased phosphorylation of p-AKT and p-GSK3β (106% ± 8% and 99% ± 6, respectively), nor to a decrease in CRMP4 phosphorylation (108% ± 8).

Together, these results show that CRMP4 is a downstream effector of Sema3E signaling, activation of the pathway leads to a decrease in its phosphorylation on Thr509. Furthermore, depletion of membrane cholesterol blocks activation of the Akt/GSK3β pathway downstream of Sema3E stimulation.

## CRMP4's cytoskeleton-binding domain is required for Sema3E signaling

The phosphorylation sites of CRMPs regulated by GSK3β are located in the C-terminal region of these proteins, at the interaction sites with the cytoskeleton (*Figure 10A*). Cdk5 and DYRK2 phosphorylate Serine 522 in CRMP4, priming it for subsequent phosphorylation on Serine 518, Threonine 509, and Threonine 514 residues by GSK3β. Sema3A has been shown to trigger CRMP2 phosphorylation at these sites, resulting in a disruption of its interaction with microtubules to allow axon repulsion (*Fukata et al., 2002*; *Uchida et al., 2005*). Our data show that axonal stimulation of subicular neurons with Sema3E is associated with a subsequent decrease in CRMP4 phosphorylation (*Figure 9C*), suggesting that the unphosphorylated form of CRMP4 transduces the signal downstream of Sema3E.

To assess this possibility, CRMP4-KO subicular neurons were transfected with plasmids encoding a non-phosphorylatable CRMP4 (CRMP4-4A) or a phosphomimetic CRMP4 for priming and GSK3β sites (CRMP4-4D). These cells were then used to evaluate the impact of these mutations following Sema3E stimulation (*Figure 10A*). Surprisingly, both mutant forms of CRMP4 restored the axonal growth induced by addition of Sema3E (*Figure 10B*, 134% and 141% of control, respectively, p < 0.0001 in both cases).

To further investigate the possible relationship between the cytoskeletal properties of CRMP4 and Sema3E-induced axonal growth, we tested a CRMP4 construct lacking its cytoskeleton-binding domain (*Figure 10A*, CRMP4ΔCyto). This domain corresponds to the last 103 residues of the protein, encompassing the priming site and the GSK3β phosphorylation sites (*Khazaei et al., 2014*). CRMP4Δ-Cyto no longer restored Sema3E-induced axonal growth (*Figure 10C*, 108% of control, p = 0.1985). These results indicate that, upon Sema3E stimulation, the cytoskeleton-binding domain of CRMP4 is required to promote axonal growth.

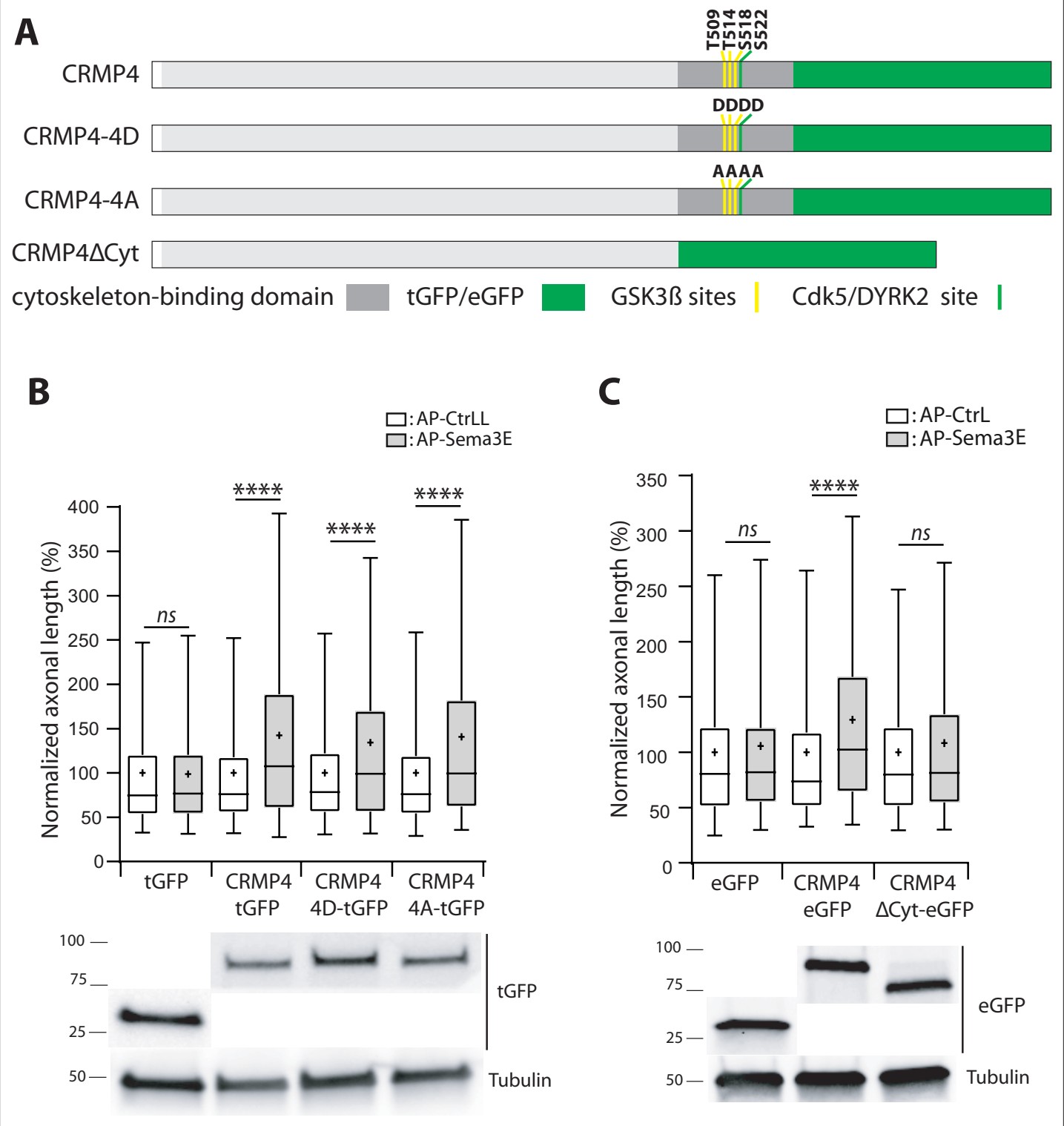

**Figure 10.** The cytoskeleton-binding domain of collapsin response mediator protein 4 (CRMP4) is required for Semaphorin-3E (Sema3E) growth-promoting activity. (**A**) Schematic representations of tGFP- or eGFP-tagged CRMP4 proteins used for rescue experiments in subicular neurons. Cytoskeleton-binding domain (grey rectangle) and the positions of the GSK3β (yellow stripes) and Cdk5/DYRK2 (green stripes) phosphorylation sites are indicated. (**B**) Quantification of the relative axonal length in CRMP4-KO subicular neurons electroporated with plasmids encoding tGFP, CRMP4-tGFP, CRMP4-tGFP non-phosphorylatable mutant (CRMP4-4A-tGFP), or CRMP4-tGFP phospho-mimetic mutant (CRMP4-4D-tGFP), in the absence (AP-CtrL) or presence of 5 nM Sema3E (AP-Sema3E). Expression levels of each CRMP protein (lower panels). (**C**) Axonal lengths measured for cultured CRMP4-knockout (KO) subicular neurons electroporated with plasmids encoding eGFP, CRMP4-eGFP, or CRMP4-eGFP mutant lacking the cytoskeleton-

*Figure 10 continued on next page*

*Figure 10 continued*

binding domain (CRMP4ΔCyt-eGFP), in the absence (AP-CtrL) or presence of 5 nM Sema3E (AP-Sema3E). Expression levels for each CRMP protein (lower panels). In B and C, crosses represent the mean, black bars represent the median, boxes represent the 25th and 75th percentiles, and whiskers represent the 5th and 95th percentiles of values from two independent cultures (**B**) or three independent cultures (**C**). Neurons (n = 300 per condition) were randomly selected; for one of the three KO-EGFP CtrL cultures, only 277 neurons could be selected. Values were normalized to 100% for values obtained in control conditions (AP-CtrL). Kruskal-Wallis non-parametric test followed by Dunn's multiple comparisons, ****p < 0.0001 and n.s. p > 0.05.

The online version of this article includes the following source data for figure 10:

**Source data 1.** Extended view of panels B-C and raw data of panels B-C.

## Discussion

In this study, we characterized CRMP4 as a new specific effector of Sema3E-mediated axonal growth. This pathway is necessary for the development of the fornix, an essential brain structure. We have previously proposed that MAP6 acts as a scaffold protein within subicular neurons, facilitating the interaction of Sema3E pathway receptors and downstream SH3-containing actors (*Deloulme et al., 2015*). Building on this knowledge, we identified CRMPs as additional effector candidates, thanks to their interactions with MAP6.

Here, we showed that CRMP4 knockdown suppresses axonal growth in subicular neurons in response to Sema3E stimulation. Axons from these cells mostly constitute the post-commissural part of the fornix. In addition, CRMP4-KO mice had defects in fornix development that were identical to those reported in Sema3E-KO mice (*Chauvet et al., 2007*). Finally, analysis of the fornix in *Crmp4/Sema3E* double heterozygote animals compared to single heterozygote animals revealed that *Crmp4* and *Sema3E* cooperate to allow the correct establishment of the fornix.

Taken together, these findings identified CRMP4 as a new effector of Sema3E-dependent axonal outgrowth implicated in fornix development. The involvement of CRMP4 in neurite outgrowth mechanisms has also been reported in the Sema3A pathway in another hippocampal neuron subtype. Thus, Sema3A stimulates the development of dendrites in a CRMP4-dependent manner (*Niisato et al., 2012*). In addition, *Crmp4* and *Sema3d* genetically interact to promote axonal growth from some peripheral neuron populations in zebrafish (*Tanaka et al., 2011*). These observations suggest a general role for CRMP4 in the attractive function of semaphorins. This property appears to be shared by other CRMPs, as both CRMP4 and CRMP1 are required to orient cerebral pyramidal neuron dendrites, and to position Purkinje cells in the developing mouse brain (*Takaya et al., 2017*; *Yamazaki et al., 2020*). Similarly, CRMP2 can rescue the inhibitory effects of CRMP4 knockdown on axonal development in cultured hippocampal neurons (*Tan et al., 2015*). Thus, CRMP1, CRMP2, and CRMP4 share some cellular functions and may have redundant roles that could be related to their extensive sequence homology, which ranges from 74% to 76% (*Charrier et al., 2003*).

However, our results show that downregulation of CRMP1 or CRMP2 does not affect the growth-promoting activity of Sema3E. These results suggest that CRMP4 plays a specific role in this signaling pathway. Interestingly, we found that knockdown of CRMP4 using siRNA stimulated basal axonal growth in cultured subicular neurons. This observation suggests that CRMP4 alone can repress axonal outgrowth in these particular neurons. Conversely, other authors report that CRMP4 promotes growth of axons as well as dendrites from neurons in the hippocampus or olfactory bulb (*Cha et al., 2016*; *Cole et al., 2004*; *Khazaei et al., 2014*; *Tan et al., 2015*; *Tsutiya et al., 2016*). Our research thus contributes to a growing body of work proposing that CRMP4 function is regulated by the molecular context of the neurons where it is expressed (*Alabed et al., 2007*; *Duplan et al., 2010*; *Girouard et al., 2020*; *Niisato et al., 2012*). Alternatively, it has been suggested that the antagonistic roles of CRMP4 observed in distinct neuronal populations could be due to differences in its phosphorylation state (*Tanaka et al., 2012*).

CRMP4-KO mice display dendritic arborization defects in hippocampal CA1 and cortical pyramidal neurons, mispositioning of Purkinje cells, and olfactory bulb lamination defects (*Takaya et al., 2017*; *Tsutiya et al., 2016*; *Yamazaki et al., 2020*). Here, we additionally show that a KO of CRMP4 is associated with a defect in post-commissural fornix development, leading to a reduction of this tract associated with fasciculation defects persisting into adulthood. This tract is crucial for cognitive functions, and in line with this role, recent investigations of white matter microstructure by diffusion tensor imaging across psychiatric disorders have highlighted alterations of the fornix in patients suffering

from both schizophrenia and autism spectrum disorders (ASD) (**Koshiyama et al., 2020**). Moreover, a de novo variant of *Crmp4* was recently identified in an adult males with ASD, and CRMP4-KO mice show a decrease in social interaction as well as a severe alterations to their sensory responses (**Tsutiya et al., 2017**). It is therefore possible that the fornix alterations we observed in CRMP4-KO mice here may contribute to these behavioral defects.

Through this study, we demonstrated that CRMP4 interacted with the Sema3E tripartite receptor complex, specifically interacting with Nrp1 and PlxD1. Complex formation between CRMP1, CRMP2, and CRMP4, and PlxA1 – a Sema3A co-receptor – has been known for several years (**Deo et al., 2004**; **Schmidt et al., 2008**). More recently, a study showed that the direct interaction of CRMP2 with PlxA3 was involved in dendritic growth under the control of Sema3A (**Jiang et al., 2020**). Binding of CRMP2 to the Ngr1 receptor and PlxA2 induced by Nogo-A was also demonstrated to limit axonal regrowth after injury (**Sekine et al., 2019**). In our study, we demonstrated that CRMP4 and Sema3E receptor interaction occurs only in DRM domains, despite the fact that these domains contain only small amounts of these proteins.

In this context, DRM is used as a broad term for a family of membrane domains resistant to high detergent concentrations, which includes lipid rafts. Thus, according to the methods used (sucrose gradients after solubilization in 1% Triton X-100 and following MβCD treatment), we opted to use the broad term DRM. Here, we show that MβCD, by destabilizing DRM, inhibits the stimulation of Sema3E-dependent axonal growth and blocks downstream activation of the Akt/GSK3β signaling pathway. Our results are consistent with studies highlighting the involvement of DRM/rafts in the Sema3A signaling pathway. For example, in leukemic T cells, Sema3A/Fas-dependent apoptosis requires Nrp1 and plexin-A1 to co-localize within lipid rafts (**Moretti et al., 2008**). Lipid rafts are also involved in Sema3A-induced growth cone collapse in neurons following the recruitment of Nrp1 to the rafts (**Guirland et al., 2004**). The presence of CRMP4 in membrane domains has been implicated in the maintenance of growth cones, suggesting that this localization places CRMP4 in an environment containing specific signaling proteins and cytoskeletal elements (**Rosslenbroich et al., 2003**). Furthermore, cerebral ischemia induced by occlusion led to CRMP4 and Nrp1 recruitment to lipid rafts, which hints that these molecules play a role in mediating signal transduction in response to neuronal damage triggered by ischemia-reperfusion (**Whitehead et al., 2010**). Other effectors of the Sema3E signaling pathway, such as Akt, are involved in a myriad of cellular processes and have also been localized within DRM. Indeed, for particular cellular processes like axonal growth, Akt must localize to regions enriched in sphingolipids and cholesterol (**Gao and Zhang, 2008**). Our results support the idea that DRM/lipid rafts regulate the spatiotemporal recruitment of Sema3E-specific receptors in order to recruit ubiquitous effectors such as Akt or GSK3β to specific membrane compartments (**Averaimo et al., 2016**; **Lingwood and Simons, 2010**). Sema3E axonal growth-promoting activity relies on the PI3K/Akt pathway, as it leads to downstream phosphorylation-induced inactivation of GSK3β (**Bellon et al., 2010**). In line with this hypothesis, we also observed that Sema3E increased phosphorylation of both Akt and GSK3β, to produce an inactive form of the latter, and thus decreased CRMP4 phosphorylation on Thr509, one of the three GSK3β phosphorylation sites (**Cole et al., 2004**). Conversely, Sema3A's repulsive activity was associated with sequential phosphorylation of CRMP2 by Cdk5 and then by GSK3β, blocking interaction between CRMP2 and microtubules in DRG-derived neurons (**Uchida et al., 2005**).

Therefore, GSK3β-dependent phosphorylation of CRMPs can be considered to be a mechanism regulating CRMP activity in response to the attractive or repulsive effects of the guidance molecules. However, results from our rescue experiments in CRMP4-KO neurons tend to suggest that the phosphorylation state of CRMP4 has little direct impact on axonal growth in response to Sema3E. Indeed, CRMP4-4A and CRMP4-4D constructs both restored the response to Sema3E in CRMP4-KO neurons. One explanation for this result could be that CRMP4 phosphorylation alters its localization rather than its cytoskeleton modulatory function. This phosphorylation-dependent localization could have been bypassed following electroporation of CRMP4 plasmids, resulting in ubiquitous overexpression of CRMP4 mutants. This hypothesis has previously been proposed for the response to Sema3D (**Tanaka et al., 2012**). In contrast, the construct deleted for the region comprising the cytoskeleton domain (CRMP4ΔCyt) did not allow recovery of axonal-growth stimulation by Sema3E, reflecting the importance of the regulatory role played by CRMP4 on the cytoskeleton.

In summary, we propose that CRMP4 is the terminal downstream effector of Sema3E in axonal-growth induction, modulating cytoskeletal dynamics to mediate axonal growth. The results presented here thus identify CRMP4 as a major new player in the growth-promoting activity of Sema3E, a signaling pathway that is much less well characterized than the repulsive role played by Sema3E in axon guidance.

## Materials and methods

### Animals

The study protocol was approved by the local animal welfare committee (Comité Local GIN, C2EA-04 – APAFIS number 8303–2016060110523424) and complied with EU guidelines (directive 2010/63/EU). Every precaution was taken to minimize the number of animals used and stress to animals during experiments.

All mouse lines used in this study were on a C57BL6 genetic background. CRMP4-KO (*Khazaei et al., 2014*) and their WT littermates were obtained by crossing CRMP4-heterozygous mice. Single and double CRMP4/Sema3E heterozygotes and their WT littermates were obtained by crossing CRMP4-heterozygote with Sema3E-heterozygote mice. Thy1-eYFP line H mice (*Feng et al., 2000*) were obtained from Jackson Labs (B6.Cg-Tg(Thy1-YFP) HJrs/J, RRID:IMSR_JAX:003782) and crossed with heterozygous CRMP4 mice to generate the heterozygous Thy1-eYFP CRMP4 mouse colony. These mice were then crossed with heterozygous CRMP4 mice to generate WT/ and KO/Thy1-eYFP-H littermate mice in inbred F1. Male and/or female mice were used from E17.5 prenatal day up to 4 months of age, depending on the experiments.

### Antibodies

The following primary antibodies were used: rabbit pan-MAP6 Ab 1:1000 (23N; homemade according to *Guillaud et al., 1998*, 1:2000 in WB), mouse MAP6-N mAb (mAb-175, homemade as described in *Pirollet et al., 1989*, 1:1000 in WB), rabbit pan-CRMPs Ab directed against the C-terminal end of CRMP2 (RIVAPPGGRANITSLG) and recognizing CRMP1, -2, and -4 isoforms (unpublished data) (C-ter; homemade, 1:5000 in WB), rabbit anti-CRMP1 (HPA035640, Sigma, RRID:AB_10670838), mouse anti-CRMP2 (C4G, IBL-America, Minneapolis, MN, RRDI:AB_1630812, 1:500 in WE), rabbit CRMP4 Ab (TUC-4, Millipore, Molsheim, France, RRID:AB_91876) (1:4000 in WB; 1:1000 in IF), rabbit phospho-CRMP4 (Thr509, homemade according to *Alabed et al., 2010*, 1:5000 in WB), goat Nrp1 Ab (AF566, Bio-techne, Noyal-Châtillon-Sur-Seiche, France, RRID:AB_355445, 1:2000 in WB, 1:800 for IF), goat PlexD1 Ab (AF4160, Bio-techne, RRID:AB_2237261, 1:2000 in WB), rabbit turbo GFP Ab (PA5-22688, ThermoFisher, Lyon, France, RRID:AB_2540616) (1:5000 in WB), rabbit VEGFR2 mAb (2479, Cell Signaling Technology, Ozyme, Saint-Cyr-l'Ecole, France, RRID:AB_2212507) (1:2000 in WB), mouse flotillin Ab (610384, BD Biosciences, Pont de Claix, France, RRID:AB_397767) (1:2000 in WB), rabbit Akt pan mAb (4685, Cell Signaling Technology, RRID:AB_2225340) (1:5000 for WB), rabbit phospho-Akt Ab (Ser473, 9271, Cell Signaling Technology, RRID:AB_329825) (1:5000 in WB), rabbit GSK-3 α/ß mAb (5676, Cell Signaling Technology, AB_10547140) (1:2000 for both WB and IF), rabbit phospho-GSK-3 α/β (Ser21/9) Ab (9331, Cell Signaling Technology, RRDI:AB_329830) (1:2000 for both WB and IF), mouse α-tubulin (clone α3A1, 1:3000 in IF), rabbit anti-GAPDH (G9545, Sigma, RRID:AB_796208) (1:6000 in WE). All secondary antibodies were purchased from Jackson Immuno Research (Interchim, Montluçon, France).

### Cell lines

HEK293T/17 were obtained from the ATCC (RRID:CVCL_4V93), cultured in DMEM medium, 10% FBS and 1% penicillin/streptomycin, authenticated using STR profiling and routinely checked for no mycoplasma contamination using MycoSEQ detection system (ThermoFisher, France).

### Plasmids

Plasmids coding for human VSV-PlxD1 (*Rohm et al., 2000*), rat FLAG-Nrp1 (*He and Tessier-Lavigne, 1997*), and human VEGFR2-FLAG (*Lamalice et al., 2006*) were used. The plasmid coding for human CRMP4-tGFP (aa 1–570, in pCMV6-AC-tGFP) was from Origene (Rockville, MD) (RG230865, # NM_001197294, NP_001184223). Derivations of this plasmid were CRMP-4A-tGFP and

CRMP-4D-tGFP, in which Thr509, Thr514, Ser518, and Ser522 residues were replaced by four alanine residues or four glutamate residues, respectively. The plasmid coding for mouse CRMP4-eGFP (aa 1–570 of NP_033494, in pCAG) was homemade. A derivation of this plasmid, CRMP4 ΔCyto-eGFP, was also produced following the deletion of the sequence coding for aa 468–570. pET21-based plasmids (pET21a(+), Novagen) were produced coding for mouse His-CRMP1 (aa 1–572 of NP_031791), mouse His-CRMP2 (aa 1–572 of NP_034085), and mouse His-CRMP4 (aa 1–570 of NP_033494).

## Brain fixations

Adult mice were fixed by transcardiac perfusion with a 4% formaldehyde solution as previously described (*Daoust et al., 2014*). Brains were then extracted and post-fixed in 4% formaldehyde solution overnight at 4°C. Embryos (E18.5 days) and perinatal (P0 and P2 days) pups were decapitated and their brains were dissected and then fixed by incubation in a 4% formaldehyde solution at 4°C for 4–5 hr. All brains are cryoprotected by immersion in sucrose solutions (15% then 30% at 4°C) until the brains sank and then frozen at –80°C in a dry-ice isopentane-bath (528471, Carlo Erba, Peypin, France).

## Histology and immunolabeling

The white matter of floated coronal sections was stained using the gold chloride method described by *Schmued et al., 2008*. Brain sections were mounted on SuperFrost plus slides (VWR, Fontenay-Sous-Bois, France), counterstained with Cresyl violet and then dehydrated in successive ethanol baths. Slices were mounted in DPX mounting medium. Each section was digitized (stereo microscope EZ4 HD, Leica Biosystem, Nanterre, France). Fornices and mammillary tracts were analyzed using ImageJ software at three anterior-posterior planes selected based on morphometric landmarks (*Franklin and Paxinos, 2008*). The first plane corresponds to the beginning of the dorsal hippocampus (Bregma: –0.94 mm), the second corresponds to the complete achievement of the dentate gyrus (Bregma: –1.34 mm), and the most posterior plane corresponds to the emergence of the habenular commissure (Bregma: –2.30 mm). The regions of interest (ROIs) of the fornices and the mammillo-thalamic tract were manually drawn and a threshold applied to measure tract surface cross-sections using ImageJ software (*Schindelin et al., 2012*). The fasciculation index was determined by dividing the smallest circle area including all fornix fibers by the fibers' area within the ROI.

Brain sections (40 μm) mounted on SuperFrost plus slides were immunostained. Coronal and sagittal (with a cutting angle of 30°) sections from pups were permeabilized in PBS-GTS (20012–019, Gibco, ThermoFisher, Dardilly, France), containing 0.2% gelatin (24350.262, VWR), 0.5% Triton X-100 (T9284, Sigma, Saint-Quentin-Fallavier, France), 0.05% Saponin (47036, Fluka, Sigma), for 1 hr, and extensively washed with PBS. Rabbit anti-CRMP4 and/or goat anti-Nrp1 antibodies were diluted in PBS-GTS at 1:1000 and 1:800, respectively, and incubated overnight at room temperature (RT). Brain sections were extensively washed with PBS-GTS before incubating with a mixture of secondary antibodies coupled to Alexa 488 or Cy3 for 2 hr at RT. Sections were extensively washed with PBS-GTS followed by PBS, and slices were mounted with fluorescent mounting medium (S3023, DAKO-Agilent, Santa Clara, CA).

The post-commissural length was quantified on 30° sagittal sections, and the diameters of the post-commissural fornix and mammillo-thalamic tract were measured on the same coronal section, corresponding to the onset of the anterior part of the AHA. Each section was digitized using an optical microscope (Axioskop 50, Zeiss, Oberkochen, Germany) equipped with a 20× (0.8 NA) dry objective and an EMCCD Quantum camera controlled by Metamorph software (Roper Scientific, Paray-Vieille-Poste, France). All morphometric measurements were taken using ImageJ software.

## Whole-mount immunostaining and tissue clearing

In this study, we used three tissue clearing methods: 3DISCO (*Figure 4C*, *Belle et al., 2014*), iDISCO (*Figure 5*, *Renier et al., 2014*), and a hybrid uDISCO method that we developed from three other protocols (Figure S2, *Hahn et al., 2019*; *Li et al., 2018*; *Pan et al., 2016*) (see supplementary material).

### 3DISCO

Whole brains from WT and CRMP4-KO littermate pups (E18.5-P2) were thawed and rehydrated in PBS at 4°C for 4 hr. Brains were permeabilized in PBS-GTS buffer containing 0.01% Thimerosal (T5125,

Sigma) overnight at 37°C and then incubated for 1 week at 37°C with anti-Nrp1 primary antibody (1:400) diluted in PBS-GTS buffer. After extensive washing, brains were incubated for 5 days with anti-goat Cy3-conjugated secondary antibody (1:500) in PBS-GTS. Brains were then cleared as previously described (*Belle et al., 2014*). Whole-brain images were acquired using a confocal microscope (LSM710, Zeiss) with 10× dry objective (NA: 0.3; WD: 5.5 mm) in the horizontal plane (x = 2.08; y = 2.08; z = 6.55 µm). Post-commissural fornices were segmented manually for each z position using the ImageJ Segmentation Editor plugin (*Schindelin et al., 2012*), and then 3D-reconstructed to produce illustrations.

### iDISCO

Immunolabeling and clearing of whole brain from WT, *Sema3e$^{+/-}$*, *Crmp4$^{+/-}$*, and *Crmp4$^{+/-}$Sema3e$^{+/-}$* newborn littermates were performed according to the iDISCO protocol (*Renier et al., 2014*). Nrp1 immunolabeling was processed as above, and brains were embedded in 0.5% agarose (2267.1, Roth, Karlsruhe, Germany) before applying the clearing procedure. Cleared brains were 3D-imaged on a lightsheet fluorescence microscope (Ultramicroscope II, LaVision BioTec, Bielefeld, Germany) using a 2× long working distance air objective lens (WD 6 mm with the dipping cap) and ImspectorPro software (LaVision BioTec). Acquisitions on the horizontal plane were obtained using Dynamic Focus with a constant lightsheet thickness of 4.1 µm (x = 3.02; y = 3.02; z = 3 µm). 3D volume images were generated using Imaris software (version 9.6, Bitplane) (Oxford Instrument, Gometz-la-Ville, France). Fornices and mammillary tracts were manually segmented using the 'Isoline' drawing mode (Imaris) by creating a mask.

### Neuron culture

Brains from WT and CRMP4-KO littermate embryos (E17.5) were dissected to obtain the subiculum. Pieces were dissociated and plated on poly-lysine/laminin-coated coverslips in DMEM supplemented with 10% horse serum and 1% penicillin/streptomycin. After 2 hr, the medium was replaced with Neurobasal medium supplemented with 1 mM glutamine, 1:50 B27 (Gibco), and treated with 5 nM alkaline phosphatase fused with Sema3E (AP-Sema3E). The AP-Sema3E and alkaline phosphatase controls (AP-CtrL) were produced as previously described (*Chauvet et al., 2007*). The effect of MβDC (C4555, Sigma) on Sema3E-induced axonal-growth stimulation was investigated as follows. MβCD (final concentration 0.5, 1.0, or 1.5 mM) was added to the culture medium when neurons were seeded, and incubated for 2 hr. The medium was then switched to supplemented Neurobasal medium, and neurons were treated with AP or AP-Sema3E for 24 hr before measuring axonal lengths.

In some experiments, neurons were electroporated (Amaxa Cell Line Nucleofector, Lonza Bioscience) either with several plasmid constructs coding for CRMP4 variants (100 pmol) or with distinct siRNAs (20 nM). Efficient knockdown of CRMP1, CRMP2, and CRMP4 was obtained using Stealth RNAi (Invitrogen, ThermoFisher), siRNA anti-*Crmp1* sequence 5'-CCC GGU GGC AUU GAU GUC AAC ACU U-3' (si-*Crmp1*), siRNA anti-*Crmp2* sequence 5'-CCA ACC ACU CCA GAC UUU CUC AAC U-3' (si-*Crmp2*), siRNA anti-*Crmp4* sequence 5'-CCU CUG GUG GUU AUC UGC CAG GGC A-3' (si-*Crmp4*). After 2 days in vitro (DIV), cultured cells were fixed, immunolabeled with an anti-tubulin antibody, and digitized using an optical microscope (Axioskop 50, Zeiss) equipped with a 20× (0.5 NA) dry objective (*Figure 2*) or using a slide scanner (Axio Scan Z1, Zeiss) equipped with a 20× (0.8 NA) dry objective (*Figures 9 and 10*). Axonal lengths were automatically measured for isolated neurons using the AutoNeuriteJ macro (*Boulan et al., 2020*).

### Phosphorylation patterns in cultured neurons

Subicular neurons cultured in Petri dishes (B35, Falcon) at 2 DIV were starved for 2 hr in Neurobasal medium without supplements, and were successively treated or not with 4 mM of MβCD for 30 min. Cultures were then treated with 5 nM of AP or AP-Sema3E for 11 min before harvesting. Neurons were lysed in 150 µL of Laemmli buffer containing 100 mM DTT. The phosphorylation state of Akt, GSK3β, and CRMP4 was investigated after separation on SDS-PAGE (4–15%) electrotransfer onto nitrocellulose membrane and immunoblotting (Bio-Rad, Marne-la-Coquette, France).

## Immunoaffinity chromatography and mass spectrometry-based proteomic analyses

Purified monoclonal antibodies directed against MAP6-N peptide-175 (mAb-175, 10 mg) were coupled on AminoLink coupling gel column (Thermo Scientific, Illkirch, France) following the manufacturer's instructions.

Twenty WT adult brains were lysed in 8 mL of PEM buffer (100 mM PIPES adjusted to pH 6.75 with KOH; 1 mM EGTA; 1 mM MgCl$_2$ and protein inhibitor cocktail) by performing six strokes with the type A pestle and two strokes with type B pestle of a Kontes all-glass Dounce homogenizer at 4°C. Homogenates were centrifuged at 70,000 g for 10 min, and supernatants were incubated with 1 mL of mAb-175-aminolink gel for 3 min at 4°C with gentle agitation. mAb-175-aminolink gel was extensively washed with PEM buffer containing 50 mM NaCl before eluting bound proteins with 4 mL of peptide-175 (1 mM in PEM buffer). Proteins from more concentrated eluted fractions were partially separated on an SDS-PAGE gel (NuPAGE 4–12%, Thermo Fisher Scientific) and stained with Coomassie blue (R250, Bio-Rad). Migration lanes were cut into 21 bands, and proteins were oxidized as previously described (*Jaquinod et al., 2012*) before in-gel digestion with modified trypsin (sequencing grade, Promega, Charbonnieres-Les-Bains, France) (*Casabona et al., 2013*).

Resulting peptides of each band were sequentially analyzed by online nanoLC-MS/MS (UltiMate 3000 and LTQ-Orbitrap Velos, Thermo Scientific). For this, peptides were sampled on a 300 µm × 5 mm PepMap C18 precolumn (Thermo Scientific) and separated on a homemade 75 µm × 150 mm C18 column (Gemini C18, Phenomenex). MS and MS/MS data were acquired using Xcalibur (Thermo Scientific).

Mascot Distiller (Matrix Science) was used to produce mgf files before identification of peptides and proteins using Mascot (version 2.7) through concomitant searches against Uniprot (Mammalia, March 2021 version; taxon *Mus musculus*), classical contaminants database (homemade) and the corresponding reversed databases. The Proline software (Bouyssie D et al., Bioinformatics, 2020) was used to filter the results (conservation of rank one peptides, minimum peptide score of 25, and minimum of two peptides including one specific peptide per protein group identified).

## Expression of recombinant proteins and pull-down

For recombinant production of CRMP proteins, *Escherichia coli* BL21(DE3) pLysS (Fisher Scientific, Illkirch, France) were transformed by pET21-*Crmp*s plasmids and protein expression was induced with 1 mM isopropyl-1-thio-β-D-galactopyranoside for 6 hr at 37°C. Bacteria were harvested by centrifugation at 5000 g (10 min at 4°C). The pellet was resuspended in 40 mM Tris-HCl pH 7.5, 300 mM NaCl, 0.1% Triton X-100, 6 mM imidazole with protease inhibitor cocktail (04693132001, Merck-Roche, Lyon, France) and lysed by three successive freezing and thawing cycles combined with sonication (3 × 10 s bursts). After centrifugation (200,000 g, 40 min at 4°C), the supernatant was added to a cobalt resin column (Fisher Scientific, 10 mL of lysate to 1 mL of beads for 1 hr at 4°C) and washed with lysis buffer. The His-tagged CRMP proteins were eluted with 40 mM Tris-HCl pH 7.5, 300 mM NaCl, 200 mM imidazole with protease inhibitor cocktail and dialyzed against 100 mM NaHCO$_3$ pH 8.3, 500 mM NaCl. Recombinant CRMPs were coupled to CNBr-Sepharose (17-0430-01, Merck-Roche, 1.4 mg of protein per milliliter of beads) as described in the manufacturer's protocol. The MAP6-N-His recombinant protein was produced in baculovirus and purified as previously described (*Cuveillier et al., 2020*).

For pull-down experiments, recombinant MAP6-N-His (2 µg) was incubated with 5 µL of control-Sepharose or CRMP-Sepharose beads in PEM buffer (100 mM PIPES, 1 mM EGTA, 1 mM MgCl$_2$, 100 mM KCl, 0.05% Tween 20, pH 7.5) in the presence of protease inhibitors for 20 min at RT. Beads were washed three times with PEM buffer and resuspended in 40 µL Laemmli buffer containing 100 mM DTT. Recovered proteins were analyzed by Coomassie staining after SDS-PAGE.

## Co-immunoprecipitation

For brain immunoprecipitation experiments, adult brains were homogenized with lysis buffer (20 mM Tris-HCl pH 7.2, 1 mM EGTA, 1 mM EDTA, 5 mM NaF, 1 mM DTT, 0.27 M sucrose, 0.5% Triton X-100) in the presence of protease inhibitor cocktail. After mechanical lysis using ceramic beads (Precellys, VWR) shaking and centrifugation of the cell lysate at 12,000 g for 20 min at 4°C, the supernatant was diluted 3× with buffer C (10 mM Tris-HCl pH 7.5, 2% glycerol) and incubated for 4 hr at RT with 20 µg of MAP6antibody (mAb175). For immunoprecipitation on cells, HEK293T/17 cells were transfected

with plasmids encoding CRMP4-tGFP, VEGFR2-FLAG, VSV-PlxD1, and FLAG-Nrp1 using a calcium phosphate transfection kit (631312, LifeTechnologies, ThermoFisher). One day after transfection, cells were lysed as described previously for brains. To test the detergent resistance of the interaction between CRMP4 and the receptors, a lysis buffer containing 1% instead of 0.5% of Triton X-100 was used. The supernatant was incubated for 1 hr at 4°C with an anti-tGFP, anti-CRMP4 (TUC4) or anti-VEGFR2 antibody, and then in the presence of Dynabeads Protein G (10,004D, Invitrogen, ThermoFisher). Beads were washed with a buffer containing 20 mM Tris-HCl pH 7.2, 1 mM EDTA, 500 mM NaCl, and 0.2% NP40. Proteins were then resuspended in 20 µL of Laemmli buffer containing 100 mM DTT and analyzed by western blot as described above.

## Sucrose-density gradient fractionation

For sucrose-density gradient fractionation, five cultures of HEK293T/17 cells in individual Petri dishes (B100, Falcon, Dutscher, Bernolsheim, France) were transfected with *Crmp4*-tGFP, VSV-*PLXND1*, and FLAG-*Nrp1* cDNAs. After 24 hr, cultures were left untreated or treated with 10 mM MβCD and then lysed in 50 mM HEPES pH 7.4, 150 mM NaCl, 5 mM EDTA, in the presence of protease inhibitors. The homogenate was vortexed for 1 hr in the presence of 1% Triton X-100 at 4°C then centrifuged at 12,000 *g* for 20 min at 4°C. Supernatant was recovered (1.5 mL) and adjusted to 40% sucrose by adding 1.5 mL of 80% sucrose in lysis buffer, placed under a 5–38% discontinuous sucrose gradient and centrifuged at 274,000 *g* for 16 hr at 4°C in Beckman SW-41Ti rotor. Fractions (12 × 1 mL each) were harvested from the top of the tube.

## Experimental design and statistical analyses

E17.5 CRMP4-KO and WT littermate embryos were obtained by crossing CRMP4-heterozygous mice. WT and KO were distinguished from heterozygote littermates based on X-Gal staining intensity of the midbrain and hindbrain. Subicula from each genotype (WT or CRMP4-KO) were pooled and the genotype of each embryo was confirmed by tail-PCR. Sema3E growth-promoting activity on siRNA-treated neurons or CRMP4-KO neurons (*Figure 2*) was analyzed on three independent cultures using 75 randomly selected neurons for each condition (two coverslips per condition). Analyses of Sema3E growth-promoting activity on MβCD-treated neurons were performed on two independent cultures using 300 randomly selected neurons for each condition (two coverslips per condition) (*Figure 9A–B*). Sema3E growth-promoting activity on rescued neurons was analyzed on two to three independent cultures, as indicated in the figure legend, using 300 randomly selected neurons for each condition (two coverslips per condition) (*Figure 10*). The box-and-whisker plots show the normalized axonal length (in %) compared to the siRNA control (*Figure 2B*) or to the AP control (AP-CtrL) condition (*Figure 2D, E and F*). Statistical comparison between AP-CtrL and AP-Sema3E-treated neuron axonal lengths was based on a Kruskal-Wallis non-parametric test followed by Dunn's multiple comparisons.

P0 *Crmp4*$^{+/-}$*Sema3e*$^{+/-}$, *Sema3e*$^{+/-}$, *Crmp4*$^{+/-}$, and WT littermates were obtained by crossing CRMP4-heterozygote animals with Sema3E-heterozygote mice. Statistical comparison between different genotypes and WT was based on a Kruskal-Wallis non-parametric test followed by Dunn's multiple comparisons.

## Statistical Analyses

All statistical analyses were performed using Prism 9.0 software (GraphPad Software), applying the tests indicated in each figure. Researchers were blind to the genotypes during data collection and analysis.

## Acknowledgements

The authors thank O Valiron for initial experiments on MAP6 neuronal partners; S Andrieu, F Mehr, L Romain, and F Rimet for animal care; F Vossier, C Corrao, L De Macedo, and N Chaumontel for technical support; C Dominici for her advice and expertise in the use of Imaris; Thomas Brown for his comments and manuscript corrections. CR and JCD particularly thank I Arnal for the current financial and logistical support. This work was supported by INSERM, CEA, University Grenoble Alpes and in part by funds awarded by the French Agence Nationale de la Recherche to AA (2010- Blanc-120201 CBioS and 2017-CE11-0026 MAMAs).

## Additional information

### Funding

| Funder | Grant reference number | Author |
|---|---|---|
| Agence Nationale de la Recherche | 2017-CE11-0026 MAMAs | Annie Andrieux |
| Agence Nationale de la Recherche | 2010-Blanc-120201 CBioS | Annie Andrieux |

The funder had no role in study design, data collection and interpretation, or the decision to submit the work for publication.

### Author contributions

Benoît Boulan, Charlotte Ravanello, Conceptualization, Data curation, Formal analysis, Investigation, Methodology, Validation, Visualization, Writing – original draft, Writing – review and editing; Amandine Peyrel, Data curation, Investigation, Writing – review and editing; Christophe Bosc, Methodology, Resources, Writing – review and editing; Christian Delphin, Alyson Fournier, Resources, Writing – review and editing; Florence Appaix, Data curation, Formal analysis, Methodology, Visualization, Writing – review and editing; Eric Denarier, Data curation, Methodology, Software, Writing – review and editing; Alexandra Kraut, Data curation, Methodology, Resources, Writing – review and editing; Muriel Jacquier-Sarlin, Conceptualization, Data curation, Methodology, Resources, Writing – review and editing; Annie Andrieux, Conceptualization, Funding acquisition, Project administration, Supervision, Validation, Writing – review and editing; Sylvie Gory-Fauré, Jean-Christophe Deloulme, Conceptualization, Data curation, Investigation, Methodology, Project administration, Supervision, Validation, Visualization, Writing – original draft, Writing – review and editing

### Author ORCIDs

Benoît Boulan http://orcid.org/0000-0001-6793-5378
Eric Denarier http://orcid.org/0000-0002-4169-397X
Muriel Jacquier-Sarlin http://orcid.org/0000-0001-8501-7511
Annie Andrieux http://orcid.org/0000-0002-4022-6405
Sylvie Gory-Fauré http://orcid.org/0000-0001-5743-8079
Jean-Christophe Deloulme http://orcid.org/0000-0002-2234-5865

### Ethics

The study protocol was approved by the local animal welfare committee (Comité Local GIN, C2EA-04 - APAFIS number 8303-2016060110523424) and complied with EU guidelines (directive 2010/63/EU). Every precaution was taken to minimize the number of animals used and stress to animals during experiments.

### Decision letter and Author response

Decision letter https://doi.org/10.7554/eLife.70361.sa1
Author response https://doi.org/10.7554/eLife.70361.sa2

## Additional files

### Supplementary files

- Transparent reporting form
- Source data 1. Statistical analysis *Figures 2*, *4–6*, *Figures 9 and 10*, *Figure 5—figure supplement 1*, *Figure 9—figure supplement 1*.
- Supplementary file 1. Summury of MAP6 interacting proteins identified by proteomic analysis.

### Data availability

All data generated or analysed during this study are included in the manuscript and supporting files. Source data files have been provided for Figure 2, Figure 4, Figure 5, Figure 6 , Figure 7, Figure 8, Figure 9, Figure 10 and supplementary File 1.

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

# Appendix 1

## Supplemental materials and methods

### Brain lysates

After euthanasia, by decapitation for E15.5, E17.5, P0 embryos or anesthetic overdose for later stages, brains were dissected out and lysed in buffer (20 mM Tris-HCl pH 7.4, 150 mM NaCl, 1 mM EDTA, 1 mEGTA, 1 mMDTT, 1 mM $MgCl_2$, 1% Triton X-100 and protease inhibitors) by six strokes with the type A pestle and two strokes with type B pestle of a Kontes all-glass Dounce homogenizer. Protein amounts between samples were equilibrated following assessment of the protein concentration by Bradford assay. An aliquot of proteins (20 µg) for each sample was loaded onto a 10% acrylamide gel for western blotting.

### Brain section X-Gal staining

E17.5 brains were incubated in 0.2% glutaraldehyde and 2% formaldehyde in PBS for 2 hr. Brain slices (50 µm) were incubated in PBS containing 5 mM potassium ferricyanide, 5 mM potassium ferrocyanide, 2 mM magnesium chloride, and 1 mg/mL X-Gal as substrate for 3–5 hr at 30°C. Slices were dehydrated and then mounted in DPX.

## Tissue clearing

### Hybrid uDISCO

Adult half-brains from WT/Thy1-eYFP-H and CRMP4-KO/Thy1-eYFP-H mice were cleared using a protocol adapted from three published methods (*Hahn et al., 2019*; *Li et al., 2018*; *Pan et al., 2016*). Brains were first dehydrated in increasing tert-butanol concentrations (30, 50, 70, 80, 90, 100, and 100%$^{v/v}$). Each bath lasted 12 hr at 32°C, and pH was adjusted to 9.0–9.5 by adding triethylamine (0.5, 0.5, 0.5, 1, 3, 15, 60, and 60 µL TEA/15 mL, respectively) (471283, Sigma) (*Li et al., 2018*; *Pan et al., 2016*). Brains were then immersed in dichloromethane (270997, Sigma) for 60 min at RT to remove lipids. Finally, brains were cleared for 5 hr at RT in di-benzyl ether (108014, Sigma) containing 0.4% *N*-propyl-gallate (P3130, Sigma) (*Hahn et al., 2019*) and washed twice in ethyl cinnamate (112372, Sigma). All incubation steps were performed in a 15 mL tube filled with solution, under mild agitation and protected from light.

