## [Editor Report]

In this work Boulan use a combination of biochemical investigations and in vivo mouse genetic analysis and colleague and describe how Sema3E affects the fornix development through CRMP4. The manuscript is of broad interest to investigators exploring the molecular mechanisms underlying circuit assembly in the mammalian central nervous system.

---

## [Decision Letter]

**Decision letter after peer review:**

Thank you for submitting your article "CRMP4 mediates fornix development through semaphorin-3E signaling pathway" for consideration by *eLife*. Your article has been reviewed by 2 peer reviewers, and the evaluation has been overseen by a Reviewing Editor and Catherine Dulac as the Senior Editor. The reviewers have opted to remain anonymous.

Essential revisions:

(1) Clarity of writing. There are numerous instances of errors in English language grammar, syntax and usage that detract from the clarity of the manuscript. The authors may wish to consider emphasizing that the growth-promoting role of Sema signaling is much less well characterized than Sema repulsion to highlight the significance of the work.

(2) Quantification of biochemistry. There are several instances where controls and quantification are missing or incomplete. Please make to have all controls and quantifications.

(2.1) Figure 1: Inputs for purified Crmps should be shown to ensure equal amounts were used in the pulldowns.

(2.2) Figure 2: Extent of knockdown should be quantified.

(2.3) Figure 8: The westerns in this figure are not convincing, especially 8D. There is a LOT of background pull down in the conditions where there is no CRMP4 present.

(2.4) Figure 9: The effect of Sema3E on downstream phsosphorylation of targets seems pretty subtle. How many times were these experiments performed? Differences should be quantified and compared statistically.

(2.5) Figure 10: This figure compares the functions of different mutant variants of Crmp4, but it is unclear whether all of these forms of the protein are comparably expressed in these assays.

(3) MAP6 has two isoforms (MAP6-E and MAP6-N) both of which are expressed in neurons. Subicular neurons predominantly express MAP6-E isoform and it has been shown that this isoform interacts with sema3E receptors to mediate sema3E signalling for axonal growth (Deloulme et al., 2015). However, authors used antibodies for the other isoform (Sema6-N) only expressed in postmitotic neurons, to investigate map6 partners implicated in fornix development. What is the rationale of that?

(4) In CRMP4 KO mice, the fornix is markedly reduced. The authors should add the measurement of axonal growth in primary culture from CRMP4-KO subicular neuron, compared to control littermates, in addition to siRNA experiments.

(5) Neurons are difficult to transfer, authors should describe in material and methods how they performed the electroporation. Some images of these experiments should be a plus (in Figure 10).

(6) Figure 9C:

(6.1) The antibody used to detect the GSK3 phosphorylation, recognizes phosphorylation in α and β subunits. Sema3E seems to increase the phosphorylation in GSK3 α and not β.

(6.2) The protein loading control is missing in this panel.

*Reviewer #1:*

Sema3E signalling is implicated in the guidance of many axonal tracts in the central nervous system. Among them, the fornix that needs the tripartite receptor complex formed by plexin-D1 (PlxD1), neuropilin-1 (Nrp1) and vascular endothelial growth factor receptor-2 (VEGFR2) for Sema3E attractant signalling. Here, Boulan et al. investigate how Sema3E governs the fornix development through CRMP4, a known player for axon development and growth cone dynamics. They found that :

– Axonal elongation of subicular neurons induced by sema3E is through CRMP4 but no other members of the CRMP family.

– CRMP4 binds the tripartite Sema3E-receptor complex in a DRM domain-dependent manner.

– Activation of Sema3E signalling leads to a reduction of the phosphorylation of CRMP4 by GSK3B

*Reviewer #2:*

Boulan and colleagues explored the role for the Collapsin Response Mediator protein CRMP4 in Sema3E signaling in the context of the development of the fornix, a key projection in the CNS that connects the hippocampus and hypothalamus. Previous work had identified a role for Sema3E in the development of this structure; however, the downstream signaling mechanism remained obscure. The authors identified multiple CRMP proteins in a proteomics approach for proteins that could interact with Map6, a known component of the Sema3E pathway. Of the CRMP proteins identified, in vitro analysis supports a specific role for Crmp4 in the Sema3E growth promoting pathway. Analysis of the expression pattern and loss of function mutants for Crmp4 revealed that Crmp4 appears to be co-expressed with Sema3E receptors, and that the mutants show reduced volume and projections in the post commissural fornix, a phenotype with some similarity to that of Sema3E mutants. Interestingly, the authors detected genetic interactions between Sema3E and Crmp4; animals heterozygous for mutations in both genes exhibit significant deficits in fornix formation. Finally, additional biochemical analysis suggests that Sema3E signals through a specific pool of receptors located in detergent resistant membrane domains to promote axon growth through Crmp4.

While generally the data and presentation are of high quality and support the major conclusions of the paper, there are some aspects of data analysis, interpretation and quantification that need to be further substantiated.

(1) Biochemistry and in vitro axon outgrowth. While it appears that there is a specific role for CRMP4 in promoting growth in response to Sema3E, one wonders how specific this effect is. Have the authors tested whether Crmp4 also prevents the action of other growth promoting factors (e.g. Netrin, BDNF).

The authors nicely show that they can detect a complex between Crmp4 and Sema3E receptor components. It would be interesting to know whether the association between Crmp4 and these receptors is in any way ligand dependent. Does Sema3E treatment promote or inhibit this interaction?

(2) Expression Data. Crmp4 expression in vivo does appear to overlap in several regions with Nrp-1; however, the pattern of localization is diffuse and does not appear to be enriched in neurites like Nrp-1 is. In primary culture of subicular neurons, CRMP-4 looks to be enriched in axons and growth cones- it would be important to show that the staining in primary culture is specific, given the apparent discrepancy between primary neurons and IF on brains.

(3) Phenotypic Analysis. The authors do a nice job measuring the area, length and volume of the PF in mutants and suggest that the phenotype 'photocopies' that of Sema3E. Of course, they mean phenocopies, but the description of the prior work on Sema3E is incomplete and could be clarified in the text. More importantly, to what extent have the authors explored the underlying cause of the reduction in PF area, length and volume. Are subicular neurons specified and maintained in normal numbers? Do fibers that should target the PF mistarget elsewhere, or do they just fail to grow. These types of measurements were included in the work on Sema3E knockouts from Neuron 2007. It would be important to look more carefully at this.

It is puzzling to this reviewer why the authors characterize the Crmp4 KO phenotype as a developmental delay. What is the basis for this?

---

## [Author Response]

Essential revisions:(1) Clarity of writing. There are numerous instances of errors in English language grammar, syntax and usage that detract from the clarity of the manuscript.

The revised version of the manuscript has been corrected.

The authors may wish to consider emphasizing that the growth-promoting role of Sema signaling is much less well characterized than Sema repulsion to highlight the significance of the work.

We thank the reviewer for this comment and have now emphasized this point at the end of the discussion.

(2) Quantification of biochemistry. There are several instances where controls and quantification are missing or incomplete. Please make to have all controls and quantifications.(2.1) Figure 1: Inputs for purified Crmps should be shown to ensure equal amounts were used in the pulldowns.

For the pulldown experiments, recombinant CRMPs were covalently bound to sepharose beads (CnBr-activated sepharose). The quality of each CRMP protein sample used for the coupling reaction is now shown in figure 1 – supplement figure 1 (Coomassie staining). As mentioned in the M and M section, identical quantities of protein (1.4 mg/ml of beads) were used for each coupling step, all couplings were very effective as no detectable amount of protein was found in solution post coupling reaction.

Nevertheless, as we have no direct way of ensuring that the functional domains of each CRMP are similarly exposed, we agree that we cannot formally compare the avidity of the different CRMP isoforms for MAP6-N. Thus, for sake of clarity we have modified the text and now say that MAP6-N can bind directly to each of the CRMP isoforms.

(2.2) Figure 2: Extent of knockdown should be quantified.

Quantification of three independent experiments are now present in figure 2A.

(2.3) Figure 8: The westerns in this figure are not convincing, especially 8D. There is a LOT of background pull down in the conditions where there is no CRMP4 present.

We apologize for the misunderstanding, actually CRMP4 was present in the control conditions in the experiments described in Figure 8D. We now explained in the figure legends how the experiment was done and that controls (CtrL) correspond to conditions where no antibody was added to the DRM/RPM fractions. Thus, the bands observed might probably correspond to Nrp1 and PlxD1 bound to beads in DRM fractions which contain large amount of lipids. As our main point was to show that CRMP4 was higher in DRM than in RPM, we now provide, in the result section, a quantification of the proportion of PlxD1 and Nrp1 present in the CRMP4 immuno-precipitate (minus the amount present in the CtrL) in both DRM and RPM fractions. For the experiment (Figure 8 D) performed in DRM fractions, the signal for Nrp1 and for PlxD1 represent 62% and 25% of CRMP4 signal, respectively, whereas in RPM fractions the signal for Nrp1 and for PlxD1 represent 0.18% and 0.69% of CRMP4 signal, respectively. A second experiment was performed and gave similar results (higher ratio in DRM than in RPM). Altogether these differences in ratio clearly indicate a preferential interaction of CRMP4 with PlxD1 and Nrp1 receptors in the DRM fractions.

(2.4) Figure 9: The effect of Sema3E on downstream phsosphorylation of targets seems pretty subtle. How many times were these experiments performed? Differences should be quantified and compared statistically.

We performed a new set of experiments (2 independent cultures with duplicate deposits). One of the experiments is now shown in figure 9C and clearly demonstrates downstream phosphorylation and dephosphorylation of targets. Indeed, quantifications of p-AKT/AKT pan, p-GSK3β/GSK3β pan and p-CRMP4/CRMP4 ratios in percent of control are now included in the result section. All signals were normalized with the signal for vinculin on the same blot.

(2.5) Figure 10: This figure compares the functions of different mutant variants of Crmp4, but it is unclear whether all of these forms of the protein are comparably expressed in these assays.

Representative western blots showing the amount of each protein are now shown in figure

10.

(3) MAP6 has two isoforms (MAP6-E and MAP6-N) both of which are expressed in neurons. Subicular neurons predominantly express MAP6-E isoform and it has been shown that this isoform interacts with sema3E receptors to mediate sema3E signalling for axonal growth (Deloulme et al., 2015). However, authors used antibodies for the other isoform (Sema6-N) only expressed in postmitotic neurons, to investigate map6 partners implicated in fornix development. What is the rationale of that?

MAP6-N and MAP6-E are the two neuronal isoforms of MAP6s proteins. MAP6-N is the longest isoform and contains the entire sequence of MAP6-E. MAP6-E is expressed in the embryonic brain and remains expressed in adult, whereas MAP6-N expression starts at E15 and strongly increases throughout adulthood. Thus, we preferentially worked with MAP6-E when addressing the developing brain and cultured subicular neurons (Deloulme 2015), whereas we used MAP6-N when working with the adult brain. In this study a monoclonal antibody (mAb175) against MAP6-N was used on adult brain extract. Moreover, mAb175 is a homemade antibody which we can easily produce in bulk to set up an affinity column. Also, for your perusal, you can see in Author response image 1 CRMP4 pulldown experiments performed on crude brain extracts from embryonic to post-natal stages. The results clearly indicate that 1/MAP6N and MAP6-E interact with CRMP4 2/ MAP6-N expression starts around embryonic day 15 making its contribution to post-commissural fornix formation plausible.

**Author response image 1. sa2fig1:** Pulldowns of MAP6-E and MAP6-N from embryonic, post-natal and adult brains using CRMP4-Sepharose (upper panel) and control-Sepharose beads (lower panel). Western blot using a pan-MAP6 antibody revealed both MAP6-N and MAP-E isoforms.

Regarding the expression of MAP6-N versus MAP6-E in post mitotic neurons, we are unable to answer definitively as both isoforms could be expressed. Accordingly, we have shown that both isoforms are expressed in cultured subicular neurons (Deloulme 2015, supplemental figure S4).

4) In CRMP4 KO mice, the fornix is markedly reduced. The authors should add the measurement of axonal growth in primary culture from CRMP4-KO subicular neuron, compared to control littermates, in addition to siRNA experiments.

Quantification was added in figure 2E and mentioned in the result section.

5) Neurons are difficult to transfer, authors should describe in material and methods how they performed the electroporation. Some images of these experiments should be a plus (in Figure 10).

Electroporation was performed using the Amaxa nucleofector device (Lonza). We classically use this technique in the lab with very good transfection efficiency (at least 60%). This information was added in the material and methods section.

Also, western blot using anti-GFP or anti-turbo GFP after transfection of the different plasmids have been added to figure 10.

For your perusal, in Author response image 2 are typical pictures showing transfected subicular neurons by electroporation with the Lonza device.

(6) Figure 9C:(6.1) The antibody used to detect the GSK3 phosphorylation, recognizes phosphorylation in α and β subunits. Sema3E seems to increase the phosphorylation in GSK3 α and not β.

We deeply apologize but the blots were inverted for p-AKT and p-GSK3α/β in the previous figure 9C. As mentioned above (point 2.4), we performed a new set of experiments that gave similar results to those found initially.

(6.2) The protein loading control is missing in this panel.

In the new experiments presented in Figure 9C, we added vinculin as a housekeeping gene.

Reviewer #2:Boulan and colleagues explored the role for the Collapsin Response Mediator protein CRMP4 in Sema3E signaling in the context of the development of the fornix, a key projection in the CNS that connects the hippocampus and hypothalamus. Previous work had identified a role for Sema3E in the development of this structure; however, the downstream signaling mechanism remained obscure. The authors identified multiple CRMP proteins in a proteomics approach for proteins that could interact with Map6, a known component of the Sema3E pathway. Of the CRMP proteins identified, in vitro analysis supports a specific role for Crmp4 in the Sema3E growth promoting pathway. Analysis of the expression pattern and loss of function mutants for Crmp4 revealed that Crmp4 appears to be co-expressed with Sema3E receptors, and that the mutants show reduced volume and projections in the post commissural fornix, a phenotype with some similarity to that of Sema3E mutants. Interestingly, the authors detected genetic interactions between Sema3E and Crmp4; animals heterozygous for mutations in both genes exhibit significant deficits in fornix formation. Finally, additional biochemical analysis suggests that Sema3E signals through a specific pool of receptors located in detergent resistant membrane domains to promote axon growth through Crmp4.While generally the data and presentation are of high quality and support the major conclusions of the paper, there are some aspects of data analysis, interpretation and quantification that need to be further substantiated.1) Biochemistry and in vitro axon outgrowth. While it appears that there is a specific role for CRMP4 in promoting growth in response to Sema3E, one wonders how specific this effect is. Have the authors tested whether Crmp4 also prevents the action of other growth promoting factors (e.g. Netrin, BDNF).

We thank the reviewer for the suggestion. It is an interesting question and accordingly we have investigated the effect of netrin and BDNF subicular neurons as shown in Author response image 3 and found no significant effect on growth. We thus did not assay the possible implication of CRMP4.

**Author response image 3. sa2fig3:** 

The authors nicely show that they can detect a complex between Crmp4 and Sema3E receptor components. It would be interesting to know whether the association between Crmp4 and these receptors is in any way ligand dependent. Does Sema3E treatment promote or inhibit this interaction?

We have performed CRMP4 immunoprecipitation experiments in HEK cells ectopically expressing the tripartite receptor complex and CRMP4 in the presence and absence of Sema3E, however, we did not see any significant variations in Nrp1 and PlxD1 expression levels. In any case the variation was very slight. More physiological experiments using subicular neurons might give more consistent results, but they are very hard to develop as the amount of receptors is very low and not suitable for quantitative biochemical investigations.

2) Expression Data. Crmp4 expression in vivo does appear to overlap in several regions with Nrp-1; however, the pattern of localization is diffuse and does not appear to be enriched in neurites like Nrp-1 is. In primary culture of subicular neurons, CRMP-4 looks to be enriched in axons and growth cones- it would be important to show that the staining in primary culture is specific, given the apparent discrepancy between primary neurons and IF on brains.

We apologize for the confusing statements. In situ analysis on whole brain sections do not allow accurate co-localization of two independent proteins. Nrp1 was used only to outline the fornix projections and to establish the presence of CRMP4 in this tract. We have simplified the text in the result section.

Regarding the specificity for CRMP4 (as found in vivo, Figure 3 A) the commercial antibody (Tuc 4) is highly specific with no labelling of CRMP4 neurons, as shown in Author response image 4.

**Author response image 4. sa2fig4:** 

3) Phenotypic Analysis. The authors do a nice job measuring the area, length and volume of the PF in mutants and suggest that the phenotype 'photocopies' that of Sema3E. Of course, they mean phenocopies, but the description of the prior work on Sema3E is incomplete and could be clarified in the text. More importantly, to what extent have the authors explored the underlying cause of the reduction in PF area, length and volume. Are subicular neurons specified and maintained in normal numbers? Do fibers that should target the PF mistarget elsewhere, or do they just fail to grow. These types of measurements were included in the work on Sema3E knockouts from Neuron 2007. It would be important to look more carefully at this.

We compared the number of subicular neurons between WT and CRMP4 KO mice using the Thy-EYFP-H transgene which is expressed by subicular neurons. Quantification of subicular neuron density (expressing eYFP) on coronal hippocampal sections of WT and CRMP4 KO brains indicated a similar density as now presented in a supplemental figure attached to figure 5 (Figure 5 – Supplement 1).

The previous Figure 5 – Supplement 1 is now Figure 5 – Supplement 2.

We did not observe any mistargeting of the post commissural fornix, just a failure of growth associated with defasciculation. Actually, it is consistent with the observations of Chauvet (2007) on Seam3E KO brains, where no mistargets were reported for the post commissural fornix but only for corticofugal and stiatonigral projections.

It is puzzling to this reviewer why the authors characterize the Crmp4 KO phenotype as a developmental delay. What is the basis for this?

We apologize and thank the reviewer for his patience. We agree that developmental delay was not the correct term as the fornix is also abnormal in adult CRMP4 brains. We have changed the text accordingly by replacing delay by defect.